# Synergistic promotions between $CO_2$ capture and in-situ conversion on Ni-CaO composite catalyst

Bin Shao[1,5], Zhi-Qiang Wang [1,5], Xue-Qing Gong [1] ✉, Honglai Liu[1,2], Feng Qian[3], P. Hu [1,4] & Jun Hu [1] ✉

The integrated $CO_2$ capture and conversion (iCCC) technology has been booming as a promising cost-effective approach for Carbon Neutrality. However, the lack of the long-sought molecular consensus about the synergistic effect between the adsorption and in-situ catalytic reaction hinders its development. Herein, we illustrate the synergistic promotions between $CO_2$ capture and in-situ conversion through constructing the consecutive high-temperature Calcium-looping and dry reforming of methane processes. With systematic experimental measurements and density functional theory calculations, we reveal that the pathways of the reduction of carbonate and the dehydrogenation of $CH_4$ can be interactively facilitated by the participation of the intermediates produced in each process on the supported Ni−CaO composite catalyst. Specifically, the adsorptive/catalytic interface, which is controlled by balancing the loading density and size of Ni nanoparticles on porous CaO, plays an essential role in the ultra-high $CO_2$ and $CH_4$ conversions of 96.5% and 96.0% at 650 °C, respectively.

Carbon capture, utilization and storage (CCUS) technology can play a crucial role to restrain the global warming of no more than 1.5 °C above pre-industrial levels[1,2]. Among various CCUS technologies, the integrated $CO_2$ capture and conversion (iCCC) has attracted more and more interests since it shows great advantages in saving energies and costs for $CO_2$ compression and transportation involved in the conventional CCUS processes[3–5]. More significantly, when the iCCC technology is applied to the $CO_2$ capture from the high-temperature flue gas, its thermo-energy can be directly converted into the chemical energy during $CO_2$ conversion, resulting in high energy efficiency[6,7].

Intuitively, the successful application of iCCC lies in the development of dual-functional materials (DFMs), of which the adsorptive

sites and catalytic sites are intimately close to each other[8]. Then, the synergistic promotion effects between the $CO_2$ capture and in situ conversion may result in both high and stable $CO_2$ capture capacity and conversion efficiency[9,10]. Currently, the reported DFMs in most iCCC processes are CaO-based composites largely due to its excellent theoretical $CO_2$ capacity (17.8 mmol g$^{-1}$) at high temperatures[11,12]. Instead of the calcination regeneration at above 900 °C in the traditional Calcium looping (CaL)[13,14], the in situ conversion of captured $CO_2$ may significantly lower the regeneration temperature and thus overcome the bottleneck problems of high energy penalties and CaO sintering. Meanwhile, catalysts such as Rh[15], Ru[16], Ni[17], Fe[18], and Co[19] in C1 chemistry of methanation, reverse water gas shift reaction (RWGS),

[1]Key Laboratory for Advanced Materials and Joint International Research Laboratory for Precision Chemistry and Molecular Engineering, Feringa Nobel Prize Scientist Joint Research Center, Centre for Computational Chemistry and Research Institute of Industrial Catalysis, School of Chemistry and Molecular Engineering, East China University of Science and Technology, 130 Meilong Road, Shanghai 200237, China. [2]State Key Laboratory of Chemical Engineering, School of Chemical Engineering, East China University of Science and Technology, 130 Meilong Road, Shanghai 200237, China. [3]Key Laboratory of Advanced Control and Optimization for Chemical Processes of Ministry of Education, School of Information Science and Engineering, East China University of Science and Technology, 130 Meilong Road, Shanghai 200237, China. [4]School of Chemistry and Chemical Engineering, The Queen's University of Belfast, Belfast BT9 5AG, UK. [5]These authors contributed equally: Bin Shao, Zhi-Qiang Wang. ✉e-mail: xgong@ecust.edu.cn; junhu@ecust.edu.cn

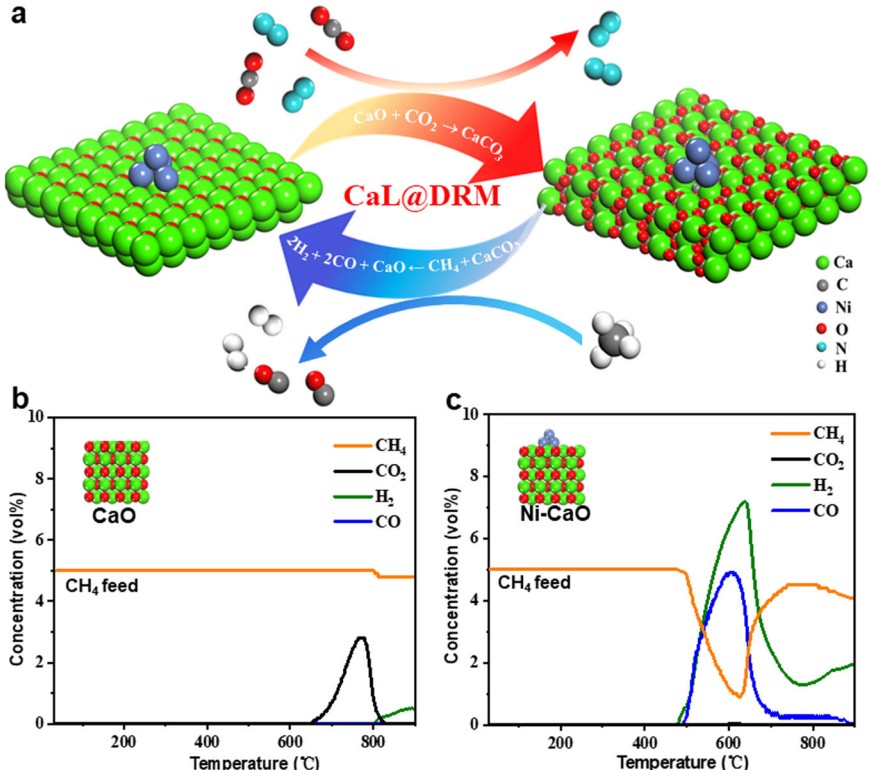

**Fig. 1 | CaL@DRM on the Ni–CaO dual-functional material. a** Schematic illustration of the proposed CaL@DRM iCCC processes. The performances of the CH₄ temperature-programmed surface reactions (TPSR) on the **b** CaO and **c** Ni–CaO-10 DFM after CO₂ pre-adsorption.

and dry reforming of methane (DRM), are usually combined within iCCC processes for the $CO_2$ conversion. However, the development of DFMs has underlined the gap in mechanism investigation, that the interactions and synergy effects between sorbents and catalysts of DFMs need significant molecular-level understanding to break through the efficiency limitation of DFMs[20]. So far, even the nature of the active sites in DFMs and its role in the conversion of captured $CO_2$ are still under debate, and it is not clear if the adsorption site can solely accommodate the $CO_2$ and its catalytic conversion can only occur at other positions[21-23], or the captured $CO_2$ can be also activated at the adsorption site for direct reaction with the incoming H species produced on the nearby catalytic sites[24-27]. Therefore, how the synergistic effects between the adsorptive and catalytic sites may work at the DFMs interface and what types of elementary steps are involved in the consecutive reaction pathways need intensive research for the development of efficient iCCC strategy[20,28].

At the same time, the dry reforming of methane (DRM) is a very useful reaction, which can convert two major greenhouse gases of $CO_2$ and $CH_4$ into valuable syngas with equimolar $H_2$ and CO. The challenges in the development of DRM mainly come from the high energy consumption due to the strong endothermic nature of this reaction and the corresponding high-temperature (>800 °C)[29]. Moreover, it also suffers from the catalyst deactivation caused by the coking[30]. Currently, most studies focus on developing efficient DRM catalysts for the activation of the two reactants, while the strategies to take advantage of their coupled reactions are still limited.

Herein, we chose the most representative high-temperature iCCC technology by integrating the Calcium-looping and dry reforming of methane (CaL@DRM) processes. For the first time, we demonstrated the synergistic promotion effects between $CO_2$ capture and in-situ conversion on the Ni–CaO DFM in a molecular way. Density functional theory (DFT) calculations of the corresponding model catalyst with a $Ni_4$ cluster on the CaO(100) substrate disclosed the synergistic reaction pathways of the direct conversion of $CaCO_3$ by the DRM, which

settles the debate about the role of captured $CO_2$ on conversion. More importantly, the mechanism of synergistic promotions provided the in-depth understanding of the long-sought consensus about the consecutive reactions involved in CaL@DRM and inspired the design and fabrication of DFMs with enriched interface between the catalytic site and adsorptive site. With the controlled loading density and the growth of Ni nanoparticles in the nanoconfined space of porous CaO substrate, such optimized Ni–CaO DFMs can give rise to high conversion efficiencies for both $CO_2$ and $CH_4$, superior to all the reported performance of conventional catalysts and DFMs for the DRM process. We anticipate that the findings will help to build an efficient way to boost the iCCC technology for Carbon Neutrality, and to shine the light on the general understanding of consecutive reactions.

## Results

The most convenient approach for the construction of possible CaL@DRM iCCC catalysts is to combine the adsorptive sites (CaO) and the catalytic sites (Ni) in one DFM (Fig. 1a). So, we intuitively fabricated a Ni–CaO-10 (Ni–CaO-$x$, $x$ denotes the loading weight percentage of Ni in sample, see Supplementary Information) DFM with Ni-rich loading considering that the activation of $CH_4$ could be particularly difficult in the in situ $CO_2$ conversion by DRM. The temperature programed surface reaction (TPSR) was first used to evaluate the CaL $CO_2$ capture process, $CH_4$ dehydrogenation and CaL@DRM processes on pure CaO and Ni–CaO-10 DFM, respectively. For the $CO_2$ capture from the simulated flue gas through the CaL process, Ni–CaO-10 exhibited similar $CO_2$-TPSR performances as CaO, giving a downward $CO_2$ adsorption peak centered at 650 °C (carbonation) and an upward $CO_2$ release peak at 850 °C (decarbonization) (Supplementary Fig. 1a, 1b)[31], demonstrating the negligible influence of supported Ni on CaL process. As the thermal cracking of $CH_4$ ($CH_4 \rightarrow C + 2H_2$, $\Delta H_{298K} = +74.6$ kJ/mol) is an endothermic reaction, it will be more favorable to occur under higher temperature. Thus, the thermal cracking of $CH_4$ was determined in this work when the temperature was high enough even

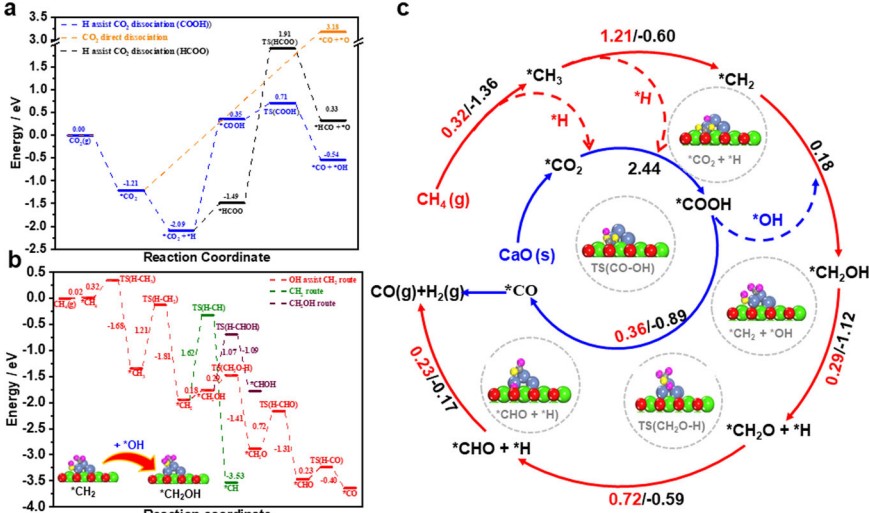

**Fig. 2 | DFT calculations and proposed reaction mechanism for CaL@DRM.**
**a** Calculated energy profiles of $CO_2$ adsorption and dissociation on the Ni$_4$–CaO(100) surface; the black dashed line represents the *H-assistant $CO_2$ dissociation of *HCOO, the orange dashed line represent the direct $CO_2$ dissociation to *CO, and the blue dashed line represents the *H-assistant *COOH pathway. **b** Calculated energy profiles of $CH_4$ dehydrogenation with the assistance of *OH species on the Ni$_4$–CaO(100) surface (red dashed line); the green dashed line represents the *CH$_2$ dissociates to *CH, and the brown dashed line represents the *CH$_2$OH dissociates to *CHOH. **c** Schematic illustration of reaction network for the CaL@DRM on the Ni$_4$–CaO(100) surface. The corresponding activation energy $E_a$ (red) and reaction energy (black) in the unit of eV for each step are also included. The optimized structures of reaction intermediates and transition state (TS) are shown in **b** and **c**. Red: O; green: Ca; blue: Ni; yellow: O of $CO_2$; Pink: H of $CH_4$. This notation is used throughout the paper.

without the help of any catalyst, giving rise to the $CH_4$–TPSR curves of the pure CaO sample and the CaO after $CO_2$ pre-adsorption (CaCO$_3$), together with the $H_2$ formation, above 800 °C (Fig. 1b and Supplementary Fig. 1c). Whereas the as-formed CaCO$_3$ caused by $CO_2$ pre-adsorption only showed a $CO_2$ desorption peak at a relatively lower temperature of 780 °C. These results demonstrate the inertness of the CaO for the DRM reaction. By contrast, the $CH_4$-TPSR measurement of the Ni–CaO-10 with pre-adsorbed $CO_2$ showed that both $CH_4$ consumption and $H_2$ production started at 485 °C (Fig. 1c), which is the same as that during the $CH_4$ dehydrogenation on the clean Ni–CaO-10 (Supplementary Fig. 1d), suggesting that the supported Ni should be the catalytically active site for CaL@DRM. In addition, the CO peak at the temperature above 500 °C can be also seen, and this slightly higher temperature over that of $CH_4$ dehydrogenation suggests the produced surface H species (*H) may even participate in the reduction of the adsorbed $CO_2$ at CaO[19,32]. Moreover, since no obvious $CO_2$ signal was observed, we may suggest that the captured $CO_2$ (CaCO$_3$) can be indeed effectively reduced by $CH_4$ to produce syngas. It is also worth mentioning that the peak values of CO and $H_2$ are centered at 600–650 °C, much lower than the reported operating temperatures of 700–800 °C of the conventional DRM[33], demonstrating the significant promotions between $CO_2$ capture at CaO and its in-situ conversion through DRM. However, we also found that after the CaL@DRM processes, the Ni–CaO-10 lost its silver-like metallic luster and appeared black in color, suggesting that the bottleneck problem of the carbon deposition in the conventional DRM may still exist[29]. Therefore, we highly anticipated that a better understanding of how the $CO_2$ adsorption and DRM conversion occur in an interactive way in our CaL@DRM processes would help guide further optimization of the Ni–CaO dual-functional materials.

### Mechanism for CaL@DRM process on Ni–CaO DFM

To illuminate the molecular mechanism of the synergistic promotions, we constructed the model Ni–CaO DFM with 4-atoms Ni clusters being supported on the CaO(100) substrate, namely Ni$_4$–CaO (Supplementary Fig. 2), and conducted the density functional theory (DFT) calculations. The calculated results showed that a single $CO_2$ prefers to be

adsorbed on the CaO surface by forming one C–O(CaO) bond and two ($CO_2$)O–Ca bonds, i.e., a carbonate-like adsorbed species (*$CO_2$) occurs (Supplementary Fig. 3). The calculated adsorption energy ($E_{ads}$) is 1.21 eV, slightly stronger than that of the $CO_2$ on the Ni$_4$ cluster ($E_{ads}$ = 1.19 eV), and similar results can be also obtained from supported Ni$_{13}$ clusters (Supplementary Fig. 4)[34,35]. Moreover, the calculated adsorption energies of $CO_2$ on the different sites of CaO(100) around the Ni$_4$ cluster are all very close with each other, indicating the moderate mobility of the adsorbed $CO_2$ on the CaO(100) surface[36,37]. Notably, the dispersion corrections by using the DFT-D2 method[38,39] and Hubbard U correction[40] were also considered in the calculation of Ni based materials, and the calculated results showed that the overall trend of $CO_2$ adsorption at different sites of the Ni$_4$–CaO(100) surface were consistent with that obtained without Grimme D2 or U corrections (Supplementary Fig. 5, 6). Then, the following conversion includes a complicated reaction network involving various intermediates produced by the evolution of $CH_4$ and adsorbed $CO_2$. For simplicity, we first calculated the energy barrier of each step of $CH_4$ dehydrogenation pathways of *$CH_4$→ *$CH_3$ + *H → *$CH_2$ + *H → *CH + *H → *C + *H on the Ni sites at Ni$_4$-CaO(100), which are 0.32 eV, 1.21 eV, 1.62 eV and 2.61 eV, respectively (Supplementary Figs. 7 and 8). The produced *H species may either combine together to produce $H_2$ or participate in $CO_2$ conversion[41–43]. According to our calculations, *H overflow from the Ni site to the nearby *$CO_2$ to form *COOH gives an energy barrier of 2.44 eV, and the as-formed *COOH can further dissociate into *CO + *OH with a small energy barrier of 0.36 eV and the overall process is exothermic by 0.89 eV (Fig. 2a and Supplementary Fig. 9). Alternatively, *H may also react with *$CO_2$ to form the *HCOO species, which needs to overcome the energy barrier of 3.40 eV to dissociate into *HCO + *O, with an endothermic reaction energy of 1.82 eV. In contrast, the direct conversion of *$CO_2$ to *CO + *O species on the Ni$_4$–CaO(100) is quite difficult, since the calculated endothermic reaction energy is as high as 4.39 eV. From calculated energetics of the above three different pathways, it can be concluded that the *COOH pathway is more favorable for the H-assisted *$CO_2$ dissociation (Fig. 2c, blue ring). These results then indicate the synergistic promotion of CaL@DRM pathway on the Ni$_4$–CaO(100) surface, as the

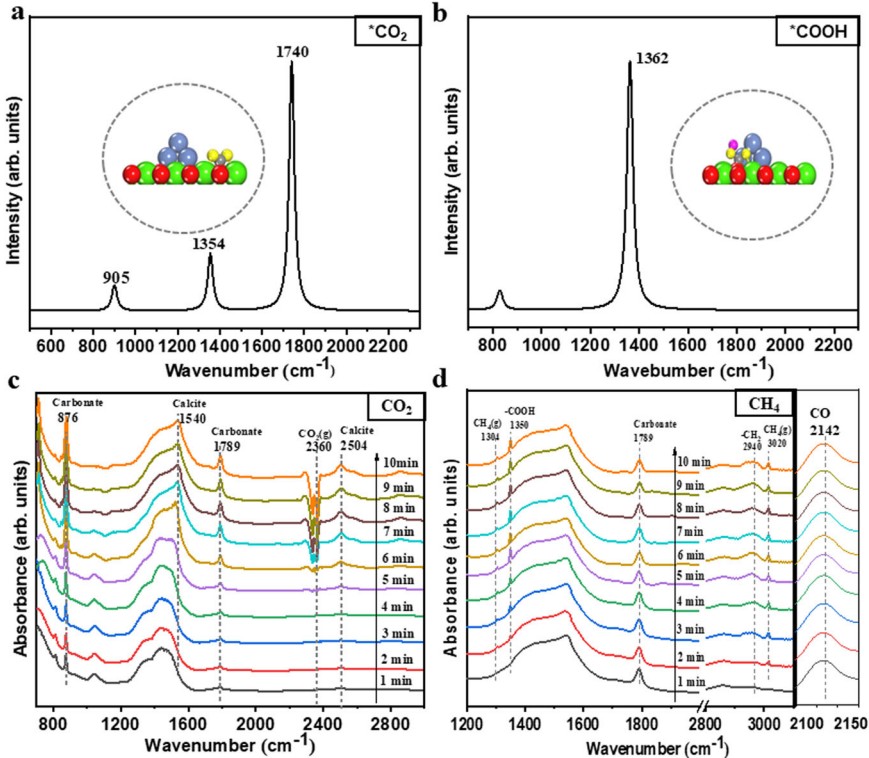

**Fig. 3 | Characteristic spectra of CaL@DRM iCCC processes on the Ni–CaO.** The DFPT calculated peaks from the asymmetric stretching of the adsorbed **a** $CO_2$ (*$CO_2$) and **b** COOH group (*COOH). In situ DRIFTS spectra during **c** the $CO_2$ adsorption stage in the atmosphere of 10 vol% $CO_2$ balanced with $N_2$, and **d** the in situ $CO_2$ conversion stage in the atmosphere of 5 vol% $CH_4$ balanced with $N_2$ of the CaL@DRM iCCC process on the Ni–CaO surface.

captured *$CO_2$ can be readily converted with the help of *H from the $CH_4$ dehydrogenation.

In addition, one may notice that in the above direct dehydrogenation of $CH_4$, rather high energy barriers need to be overcome when the intermediate *$CH_2$ species undergoes deeper dehydrogenation (Supplementary Fig. 7). Interestingly, according to our calculations, the *OH produced from the conversion of *$CO_2$ can readily react with the *$CH_x$ ($x = 2, 1, 0$) at the Ni sites to form *$CH_xOH$ (Supplementary Fig. 8, 10). It is found that *$CH_2OH$ dissociate to *CHOH species need to overcome energy barrier of 1.07 eV, which is 0.78 eV higher than that of *$CH_2OH$ dissociate to *$CH_2O$ species (Fig. 2b). So, the produced *OH would assist *$CH_2$ oxidation through the pathway *$CH_2$ + *OH → *$CH_2OH$ → *$CH_2O$ + *H → *CHO + *H → *CO + *H, with the energy barrier of 0.18 eV, 0.29 eV, 0.72 eV and 0.23 eV for each step, respectively (Fig. 2c, red ring). Therefore, the overflow of OH* produced in the H-assistant *$CO_2$ conversion can actually accelerate the $CH_4$ dehydrogenation on the $Ni_4$–CaO(100) surface, which provides another strong evidence of the synergistic promotions. It is also worth mentioning that the novel pathway of the $CH_4$ dehydrogenation disclosed here suggests the important role of the suitable interface of Ni–CaO DFM as it may stride across the carbon deposition in the conventional DRM.

We further verified this synergistic mechanism for the CaL@DRM processes through the in-situ diffuse reflectance infrared Fourier transform spectroscopy (DRIFTS) and the density-functional perturbation theory (DFPT) calculation[44–46]. In the $CO_2$ capture stage, the optimized carbonate-like adsorbed *$CO_2$ species one the CaO(100) surface obtained from DFT calculations was used, and the calculated asymmetric stretching frequencies are at 1740 and 905 $cm^{-1}$ (Fig. 3a). Correspondingly, the increasing peaks of the carbonate (*$CO_2$) on the CaO surface at 1789 and 876 $cm^{-1}$, and of the calcite ($CaCO_3$) at 1540 and 2504 $cm^{-1}$ can be clearly observed in the DRIFTS spectra (Fig. 3c)[47]. After the $N_2$ elution, the gas was switched into $CH_4$ for the following

$CO_2$ conversion stage. According to the DFPT calculation, the asymmetric stretching of the adsorbed monodentate *COOH on the CaO(100) gives the peak at 1362 $cm^{-1}$ (Fig. 3b). Accordingly, a newly formed peak at 1350 $cm^{-1}$ in the DRIFTS spectra can be also seen and verified to be the key intermediate of *COOH (Fig. 3d)[48]. These results indeed further confirmed the proposed *H-assisted pathway for the *$CO_2$ conversion. Moreover, the appearance of a symmetric vibration peak of v($CH_2$) at 2940 $cm^{-1}$ may also support the *OH-assistant $CH_4$ dehydrogenation[48]. As expected, the CO peak at 2142 $cm^{-1}$ gradually increases with the decrease of carbonate peak at 1789 $cm^{-1}$ [49–51]. Again, since no $CO_2$ peak was observed, it can be further clarified that the captured *$CO_2$ ($CaCO_3$) is indeed directly reduced into CO through the in situ CaL@DRM pathway.

### Control synthesis of Ni–CaO DFMs

From the above results and discussions, it can be expected that the fabrication of proper adsorptive/catalytic interface is essential for the synergistic promotion between $CO_2$ capture and in-situ conversion, where the reaction efficiency of the *H-assistant *$CO_2$ conversion and the *OH-assistant $CH_4$ dehydrogenation should be compatible with each other. In the current work, the Ni–CaO DFMs synthesized by a simple sol–gel method all exhibited a branched coral-like porous morphology (Fig. 4a and Supplementary Fig. 11, Supplementary Table 1). After reduction in $H_2$ atmosphere at 700 °C, the X-ray diffraction (XRD) patterns of all samples showed the successful transformation of NiO (JCPD 47-1049) to Ni (JCPD 04-0850), with the dominant CaO crystals being unchanged (JCPD 48-1467) (Fig. 4b and Supplementary Fig. 12). The most of Ni species in Ni–CaO-x are well-distributed throughout the CaO matrix (Supplementary Fig. 13). Along with the increased density of Ni nanoparticles, the average sizes of Ni nanoparticles on the Ni–CaO-2.5, Ni–CaO-5, and Ni–CaO-10 catalysts varied as $13.2 \pm 2.5$ nm, $13.4 \pm 2.4$ nm, and $27.6 \pm 3.0$ nm, respectively (Fig. 4c and Supplementary Fig. 14), suggesting there is a trade-off

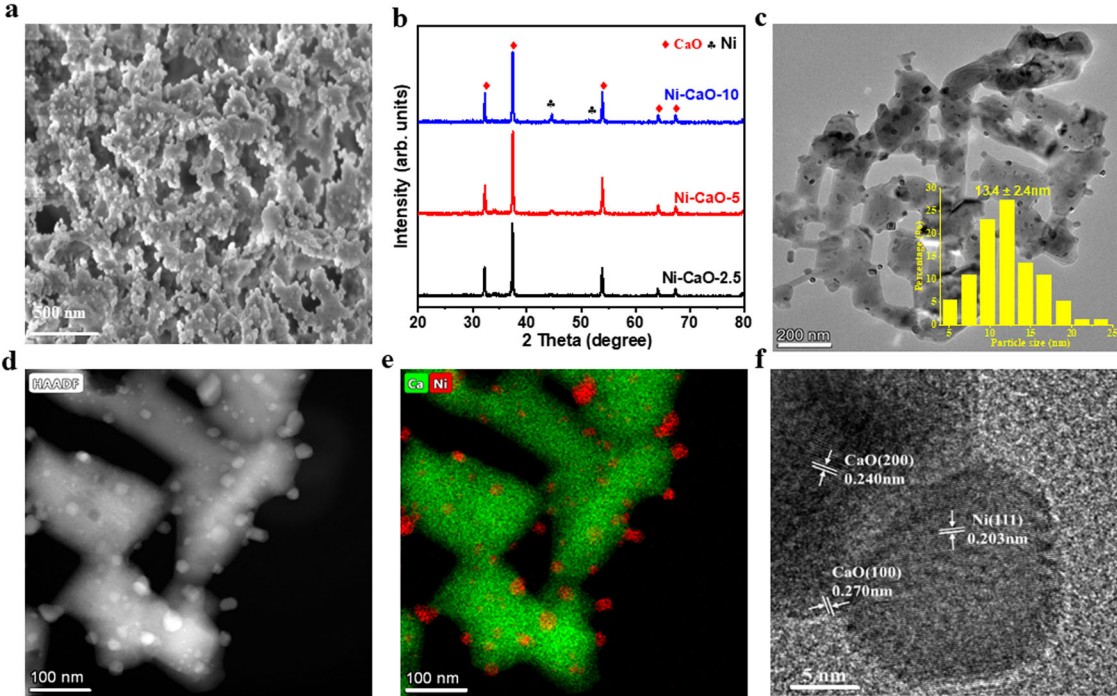

**Fig. 4 | Structural characterizations of Ni–CaO DFMs. a** Micrometer-scale morphology of Ni–CaO-5 (scale bar = 500 nm). **b** XRD patterns of Ni–CaO-2.5, Ni–CaO-5, Ni–CaO-10. **c** TEM image (scale bar = 200 nm) and size distribution of Ni nanoparticles at Ni–CaO-5. **d** STEM image of Ni–CaO-5 (scale bar = 100 nm). **e** Energy-dispersive X-ray elemental mapping images of Ni–CaO-5. **f** High-resolution TEM of Ni–CaO-5 (scale bar = 5 nm).

between the Ni loading density and the size of Ni nanoparticles to achieve the expected adsorptive/catalytic interface in DFM. We estimated the proportion of the adsorptive/catalytic interface by making a distinction between the bonded Ni toward CaO and the free one. On the XPS spectrum of Ni $2p_{3/2}$ of Ni–CaO-x, two peaks of the bonded Ni at 856 eV and the free one at about 854 eV were observed[31], giving that the bonded Ni of 67, 73, and 56% (atomic concentration) of total Ni on Ni–CaO-2.5, Ni–CaO-5, and Ni–CaO-10, respectively (Supplementary Fig. 15). Therefore, Ni–CaO-5 exhibited the largest adsorptive/catalytic interface as the sizes of the supported Ni nanoparticles kept quite small while the amount of Ni loading was large enough. The scanning transmission electron microscopy (STEM) images complemented by energy-dispersive X-ray elemental mapping (EDX) further demonstrated the homogeneous mixing of catalytic Ni sites with adsorptive CaO sites on the nanoscale (Fig. 4d, e). More specifically, the interface between the closely packed Ni nanoparticle and the CaO support was investigated by the high-resolution TEM (HRTEM) image (Fig. 4f), where the lattice spacings of 0.203 nm of Ni(111) and 0.270 nm of CaO(100) can be clearly seen[52].

## ICCC performance of Ni–CaO DFMs

Then, the iCCC performance was systematically measured on the obtained Ni–CaO-x DFMs in one fixed-bed reactor through consecutive CaL@DRM processes (Fig. 5a and Supplementary Fig. 16). At the fixed temperature of 650 °C, in the stage of $CO_2$ capture from the simulated flue gas (10 vol.% $CO_2$ balanced with $N_2$), all the Ni–CaO-x (x = 2.5, 5, 10) DFMs exhibited excellent $CO_2$ adsorption capacities (Supplementary Table 2), giving slightly decreased values of 12.10 mmol g$^{-1}$, 11.91 mmol g$^{-1}$, and 11.58 mmol g$^{-1}$ along with the increase of Ni loading (Fig. 5b). After $N_2$ purge and in the stage of in-situ conversion by switching the gas to 5 vol% $CH_4$ balanced with $N_2$, the conversions of $CH_4$ ($CO_2$) reached 86.4% (84.3%), 94.1% (90%), and 95.4% (92.7%) on these catalysts, respectively; and the determined $R_{H/C}$ in the syngas products were 0.88, 1.06, and 1.24 (Fig. 5c and Supplementary Table 3). We can clearly see that without sufficient Ni–CaO

interfaces on the Ni–CaO-2.5, $CH_4$ dehydrogenation can hardly provide enough *H to assistant the *$CO_2$ conversion. Instead, the produced $H_2$ may participate the reduction of the excessive $CO_2$ through the reversed water gas shift reaction (RWGS), resulting in the $R_{H/C}$ lower than 1. By contrast, too high loading density of Ni on the Ni–CaO-10 will also lead to the mismatching between the $CO_2$ conversion and *OH-assistant $CH_4$ dehydrogenation, resulting in the $CH_4$ dehydrogenation by itself on the Ni particles, and hence the $R_{H/C}$ larger than 1. It was further confirmed the deposited carbon formed on Ni–CaO-10 rather than Ni–CaO-5 in CaL@DRM process by Raman spectra (Supplementary Fig. 17). This can just explain why we observed in the above $CH_4$–TPSR measurements that the carbon deposition occurred when rich Ni was loaded on Ni–CaO-10. Accordingly, to improve the comparability in activity among the Ni–CaO DFMs with different Ni loadings, the measured activities of Ni–CaO-x were also calculated in terms of per gram of Ni loading, and the Ni–CaO-5 still gave the best performance, with the highest $H_2$ and CO yields of 0.182 mmol s$^{-1}$ g$_{Ni}^{-1}$ and 0.172 mmol s$^{-1}$ g$_{Ni}^{-1}$, respectively (Fig. 5c). It is worth mentioning that such $R_{H/C}$ close to unity obtained under so high conversions suggests the complete DRM, with negligible side reactions such as methane decomposition or RWGS. In contrast, for the conventional DRM process with co-feeding of $CO_2$ and $CH_4$ under the similar operating conditions, Ni–CaO-5 shows much worse activity that the conversions of $CO_2$ and $CH_4$ significantly decreased to 75.0% and 60.0% at 650 °C, respectively, and the $R_{H/C}$ is only about 0.5 (Supplementary Fig. 18), confirming the synergistic promotions between $CO_2$ capture and in situ conversion through the CaL@DRM iCCC processes.

To elaborate upon the importance of interface structures for the iCCC performance, we regulate the size of Ni nanoparticles by varying the amount of soft template at the fixed loading of Ni (5 wt%) in the sol-gel process. As the result of the different nanoconfined spaces constructed by the template, a series of Ni–CaO-5(d) DFMs with the Ni particle size (d) ranging from 8.2 to 17.2 nm were obtained (Supplementary Fig. 19). As listed in Supplementary Table 4, the dispersion degree of Ni increased with the decrease of Ni nanoparticle size, and

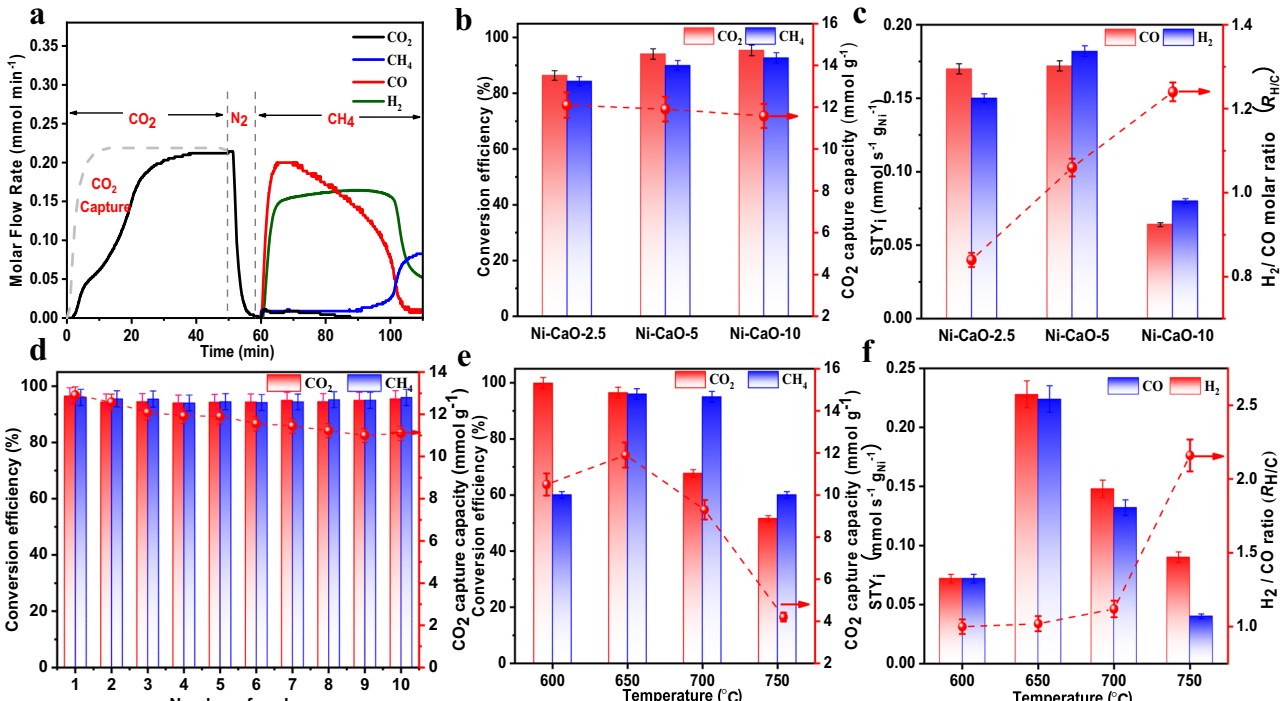

**Fig. 5 | ICCC performance of Ni–CaO DFMs. a** Molar flow rate of the effluent gas in one cycle of CaL@DRM on the Ni–CaO-5. The effects of Ni loading content on **b** average conversion of $CO_2$ and $CH_4$ along with $CO_2$ capture capacity, and **c** specific yield of $H_2$ and CO (average yield per gram of the loaded Ni) along with the $R_{H/C}$ in the syngas product at 650 °C. **d** Stability of 10 cyclic CaL@DRM iCCC processes in one Ni–CaO-5(8.2) packed reactor at 650 °C. The effects of temperature on **e** the average conversion of $CO_2$ and $CH_4$ along with the $CO_2$ capture capacity and **f** specific yield of $H_2$ and CO (average yield per gram of the loaded Ni) along with the $R_{H/C}$ in the syngas product on Ni–CaO-5(8.2). Error bars mean ± standard deviations calculated from three independent measurements.

hence, the smaller Ni nanoparticles, the larger the interface can occur at Ni/CaO. The Ni–CaO-5(d) DFMs also gave improved $CO_2$ adsorption and conversion with the decreased size of Ni nanoparticles. Specifically, Ni–CaO-5(8.2), with the smallest Ni particle size, exhibited the maximum $CO_2$ capture capacity of 12.8 mmol $g^{-1}$ and the best $CO_2$ and $CH_4$ conversion of 96.5% and 96.0%, respectively, together with the syngas production yield as high as 0.452 mmol $s^{-1}$ $g_{Ni}^{-1}$ (Supplementary Fig. 20). Such changing pattern of the iCCC performance with the Ni nanoparticle size provided further evidence that the enlarged Ni/CaO interface can facilitate the synergistic reaction pathways to promote the CaL@DRM reaction on Ni–CaO DFM. Compared with catalysts reported in other studies (Supplementary Table 5), the Ni–CaO-5(8.2) DFM produced by the modified sol–gel method showed the highest $CO_2$ conversion and $CH_4$ conversion of 96.5% and 96.0%, respectively. More significantly, the $H_2$/CO molar ratio in the elution profile was nearly 1, suggesting that Ni–CaO-5(8.2) DFM had a much better resistance to carbon deposition compared with the Ni-Ca@Zr and Ni–CaO catal@sorbent[26,53].

In addition, to realize the consecutive reactions of the exothermic $CO_2$ capture and the endothermic conversion in the same reactor for the iCCC processes, the reaction temperature plays a pivotal role[6,54]. The syngas production is determined not only by the catalytic efficiency in the in situ conversion, but also the $CO_2$ capacity in the $CO_2$ capture stage. In the temperature range of 600–750 °C, in consistence with the $CO_2$-TPSR curve, the $CO_2$ adsorption capacity of Ni–CaO-5(8.2) shows a maximum at 650 °C (Fig. 5e); while the $CO_2$ conversion efficiency decreases with increasing temperature and the $CH_4$ conversion efficiency approaches the maximum at 700 °C, and overall, the production of $H_2$ and CO shows an outstanding performance at 650 °C (Fig. 5 f). Notably, the $R_{H/C}$ booms up to 2.4 at 750 °C, suggesting the excessive temperature will lead to the more significant $CH_4$ dehydrogenation, which may even cause the coking on the Ni-based

catalysts[55]. Accordingly, 650 °C is the optimized temperature for the consecutive CaL@DRM processes. In this work, 10 successive runs at 650 °C in the Ni–CaO-5(8.2) packed reactor were conducted, and the results showed an extremely stable $CO_2$ and $CH_4$ conversions of 96.5% and 96.0%, respectively, with only a slight decrease in the cyclic $CO_2$ capture capacity (Fig. 5d). Moreover, there was no obvious carbon deposition on Ni–CaO-5(8.2) after the 10 cyclic runs (Supplementary Fig. 21), demonstrating the excellent performance of this synergistic CaL@DRM iCCC processes.

## Discussion

In summary, we demonstrated the synergistic effects between $CO_2$ capture and in-situ conversion through the consecutive CaL@DRM processes on the Ni–CaO DFM catalysts. The rich Ni–CaO interfaces with the catalytic sites and adsorptive sites intimately closed with each other promote the $CH_4$ dehydrogenation on the catalytic Ni site to form *H intermediate which can overflow to the nearby CaO surface and react with the captured *$CO_2$ there to facilitate its reduction; at the same time, the *OH intermediate produced from the *$CO_2$ conversion can reversely participate in the $CH_4$ dehydrogenation and induce the *$CH_2$ oxidation to form CO. The critical step for the formation of *COOH intermediate from $CO_2$ during the CaL@DRM processes was captured by the in-situ DRIFTS spectra, demonstrating the reliability of the proposed synergistic promotion mechanism. Furthermore, the adsorptive/catalytic interface on Ni–CaO was maximized by balancing the loading density of Ni and its aggregation growth on porous CaO. After optimizing the temperature matching between the exothermic $CO_2$ capture and endothermic conversion in one fixed-bed reactor, the extraordinary synergistic promotions of CaL@DRM performance were achieved at 650 °C that the $CO_2$ and $CH_4$ conversions were as high as 96.5% and 96.0%, respectively, with the syngas production yield of 0.452 mmol $s^{-1}$ $g_{Ni}^{-1}$. Due to the synergistic effects on facilitating the

pathways of $CaCO_3$ regeneration and $CH_4$ dehydrogenation, the bottleneck deactivation problems of the CaO sintering and carbon deposition were successfully surpassed, resulting in high recycle-stability. We anticipate that our findings of the synergistic promotions in coupling carbon capture and in-situ conversion could guide the design of highly efficient DFMs for iCCC processes.

## Methods

### Chemicals

Calcium nitrate tetrahydrate ($Ca(NO_3)_2 \cdot 4H_2O$, 99.99%), nickel nitrate hexahydrate ($Ni(NO_3)_2 \cdot 6H_2O$, 99.99%), urea($CO(NH_2)_2$, 99.99%) and citric acid monohydrate ($C_6H_{10}O_8$, 99.99%) were purchased from Sinopharm. All chemicals were in the analytical reagent-grade and used as received. Deionized water was used in all synthesis and washing processes.

### Synthesis of Ni−CaO-*x* dual function materials (DFMs)

Ni−CaO-*x* DFMs were prepared by a simple one-pot sol−gel approach and a subsequent calcination. Typically, 8.43 g of $Ca(NO_3)_2 \cdot 4H_2O$ and 7.50 g of citric acid monohydrate were dissolved in 20 mL of distilled water at room temperature under stirring for 0.5 h. Then specific amounts of $Ni(NO_3)_2 \cdot 6H_2O$ were added into the mixture under stirring for an additional 0.5 h. Then, the mixture was continuously stirred at 90 °C to form a translucent pale-green sol. The sol was transferred in an oven at 120 °C for 12 h to obtain a dried gel. The dried gel was calcined in a muffle furnace under air at 800 °C for 4 h at a temperature ramp rate of 2 °C min⁻¹ to produce the Ni−CaO-*x* dual function materials, where *x* represents the weight fraction of Ni. The DFMs were fully reduced by $H_2$ at 700 °C for 2 hours before use. In addition, pure CaO was also prepared as a reference sample according to the above-mentioned procedures.

For Ni−CaO-5(*d*) DFMs loading the same Ni loading (5 wt%), prepared by the modified sol−gel method, different molar ratio (0.25, 0.5, 1, and 2) of soft organic template (citric acid and urea) with metal precursor were dissolved in water. The other prepared steps are the same as above. But the dried gel firstly was calcined under $N_2$ at 800 °C for 2 h and then calcined under air at 600 °C for 2 h. Ni−CaO-5(*d*) samples under investigation have varying Ni mean particle sizes (*d*), as listed in Supplementary Table 4.

### Characterization

Powder X-ray diffraction (PXRD) patterns were obtained on a Bruker D8 Advance X-ray powder diffractometer equipped with a Cu sealed tube ($\lambda = 1.54$ Å) at a scan rate of 0.02 s⁻¹. Scanning electron microscopy (SEM) was conducted on a Helios G4 UC SEM-FIB (15 kV). Samples were pre-treated via Pt sputtering before the observation. Transmission electron microscopy (TEM) and Scanning transmission electron microscopy (STEM) images were performed on a Thermo-Fisher Talos F200X (FETEM, 200 kV) equipped with an energy dispersive spectrometer (EDS) for determining the elemental distribution. High angle annular dark field (HAADF)-STEM images were recorded by using a convergence semi angle of 11 mrad, and inner and outer collection angles of 59 and 200 mrad, respectively. Nitrogen adsorption-desorption isotherms were measured at 77 K using a Micromeritics ASAP-2020, from which the specific surface area and the pore size distribution of samples were calculated based on the Brunauer−Emmett−Teller (BET) model and the Barrett−Joyner−Halenda (BJH) approach, respectively. Before the measurement, the samples were degassed at 200 °C for 5 h. Elemental analysis was performed using an inductively coupled plasma atomic emission spectrometer (ICP, Varia 710 ES). X-ray photoelectron spectroscopy (XPS) of the samples were recorded on a Thermo-Scientific K-Alpha Spectrometer. All peaks were corrected by setting the C 1*s* peak of 284.6 eV as the reference. Fourier transform infrared spectra (FTIR) was performed on a NEXUS 470 spectrometer in the range of 4000−400 cm⁻¹. Prior to the measurement, the sample was mixed with potassium bromide and pressed into a wafer.

The temperature-programmed reduction in hydrogen ($H_2$-TPR) was carried out on an automated chemisorption flow analyzer (Autochem 2720, Micromeritics), equipped with a thermal conductivity detector to optimize the reduction conditions of DFMs. In a typical experiment, 50 mg of the calcined sample was loaded in a quartz reactor and heated to 300 °C in the nitrogen atmosphere (50 mL min⁻¹) for the dehydration. After the temperature was cooled down to 50 °C, the gas was switched into 10 vol% $H_2$ balanced with Ar (50 mL min⁻¹). Subsequently, the temperature was increased again with a heating rate of 10 °C min⁻¹.

The temperature programmed surface reaction (TPSR) was performed on Autochem 2720. The performance of the Calcium looping (CaL) $CO_2$ capture process, the $CH_4$ dehydrogenation, and the integrated $CO_2$ capture and conversion of CaL@DRM processes was investigated by the $CO_2$-TPSR, $CH_4$-TPSR, and $CH_4$-TPSR with $CO_2$ pre-adsorption, respectively. Prior to each test, the sample (0.1 g) was pretreated under 10 vol% $H_2$ balanced with $N_2$ at 700 °C for 30 min. For the CaL process determined by the $CO_2$-TPSR test, samples were heated from 50 to 900 °C with a heating rate of 10 °C min⁻¹ under the gas flow of 10 vol% $CO_2$ balanced with $N_2$ at a rate of 50 ml min⁻¹. For the $CH_4$ dehydrogenation process determined by the $CH_4$-TPSR test, the operation conditions are the same except the gas flow changed into 5 vol% $CH_4$ balanced with $N_2$. For the CaL@DRM processes, the sample was exposed to a gas flow of 10 vol.% $CO_2$ balanced with $N_2$ at a rate of 50 ml min⁻¹ for 30 min at 650 °C to make DFMs saturated with $CO_2$. Then the sample was cooled down to 50 °C. Afterwards, the $CH_4$-TPSR test was carried out under the same conditions as above. The nondispersive infrared analyzer (Smart Pro, Shandon) was used to monitor the concentration changes of each substance (CO, $CO_2$, $CH_4$, and $H_2$) participated during all TPSRs in effluents.

The produced intermediates during CaL@DRM process were monitored by an in situ diffuse reflectance infrared Fourier transform spectroscopy (DRIFTS, Thermo-Scientific, Nicolet 6700), equipped with a diffuse reflection attachment reaction cell (DRK-3 Praying Mantis Harrik). The spectra were recorded in the range of 3000−750 cm⁻¹ per 64 scans with a resolution of 4 cm⁻¹. Prior to each testing, the sample (~5 mg) was pre-reduced in the atmosphere of 10 vol% $H_2$ balanced with He (25 ml min⁻¹) at 700 °C for 2 h. Then, the gas was switched into $N_2$ to remove the residual $H_2$ for 30 min by decreasing the temperature to 600 °C, and the background reference signals were collected. The $CO_2$ capture was performed in the atmosphere of 10 vol% $CO_2$ balanced with $N_2$ (50 mL min⁻¹) for 10 min. After the $N_2$ elution for 8 min, the in situ $CO_2$ conversion was performed by switching the gas into 5 vol% $CH_4$ balanced with $N_2$ (50 mL min⁻¹) for another 10 min.

### $CO_2$ capture and in situ conversion test

The $CO_2$ capture and in situ conversion (CaL@DRM processes) were conducted in one fixed-bed column. The flow rates of $N_2$, $CO_2$, $H_2$, and $CH_4$ were controlled by the mass flow controllers (Horiba Metron), respectively. The products in the outlet gas were analyzed by the gas chromatography (GC, Haixin 950) with a thermal conductivity detector (TCD) and a flame ionization detector (FID). In addition, an in situ nondispersive infrared analyzer (Smart Pro, Shandon) was used to monitor the concentration changes of CO, $CH_4$, $H_2$, and $CO_2$ continuously. In a typical experiment, approximately 0.1 g DFMs was added to a quartz tube ($\Phi$10 mm × 150 mm) and packed with height of about 10 mm, then placed in the reactor furnace. The first step was the sample prereduction, which was carried out at 700 °C in the gas flow of 10 vol% $H_2$ balanced with $N_2$ at a rate of 50 ml min⁻¹ for 2 h. The second step was the $CO_2$ capture, in which the gas was switched to the simulated flue gas of 10 vol% $CO_2$ balanced with $N_2$ at 50 ml min⁻¹ (WHSV = 30 L g⁻¹ h⁻¹) and at a specific temperature such as 600, 650,

700, and 750 °C for 1 h. The third step was the purge. The pipeline was purged with pure $N_2$ for 5 min. The fourth step was the in situ conversion, in which the temperature was kept the same as in $CO_2$ capture, and the gas was switched to 5 vol% $CH_4$ balanced with $N_2$ at a flowrate of 50 ml min$^{-1}$. Moreover, we performed the blank test in the same fixed bed under the same operating conditions by using an inert material ($SiO_2$) with similar particle size to that of DFMs.

The $CO_2$ adsorption capacity ($q$) was calculated by Eq. (1)

$$q = \frac{\int_0^{ts}[F_{CO_2,in} - F_{CO_2,out}(t)]dt}{M_0} \tag{1}$$

where $F_{CO_2,in}$ is the $CO_2$ molar flow rate in the inlet gas of the simulated flue gas, $F_{CO_2,out}$ is the $CO_2$ molar flow rate in the outlet gas, $ts$ is the duration time of the capture step, and $M_0$ is the sample mass.

The $CO_2$ and $CH_4$ conversion efficiency was calculated by Eq. (2)

$$X_i = \frac{\int_0^{ts}[F_{i,in} - F_{i,out}(t)]dt}{F_{i,in} \times ts} \times 100\% \tag{2}$$

where $X_i$ (%) is the conversion of $CO_2$ or $CH_4$, $F_{i,in}$, and $F_{i,out}$, represent the molar flow rate of $CO_2$ or $CH_4$ in the inlet gas and the outlet gas, respectively. $ts$ is the duration time of the in-situ conversion step.

The average space time yield ($STY_i$) of CO or $H_2$ was calculated by Eq. (3):

$$STY_i = \frac{\int_0^{ts}[F_{i,out}(t)]dt}{M_0 \times ts} \tag{3}$$

where $F_{i,out}$ represents the molar flow rate of CO or $H_2$ in the outlet gas, $ts$ is the duration time of the in-situ conversion step, and $M_0$ is the sample mass.

The $H_2$/CO molar ratio ($R_{H/C}$) of syngas was determined by the ratio between $H_2$ and CO concentrations in the outlet gas.

For comparison, the conventional DRM test was also conducted as above, except that the $CO_2$ capture step was skipped, and the feeding gas was changed into the mixture of 10 vol% $CO_2$ and 10 vol% $CH_4$ balanced with $N_2$.

## DFT calculations

All spin-polarized DFT calculations were carried out using the Vienna Ab–Initio Simulation Package (VASP)[56,57]. The projector augmented wave (PAW) method[58] and the Perdew–Burke–Ernzerhof (PBE)[59] functional under the generalized gradient approximation (GGA)[60] were applied throughout the calculations. The kinetic energy cut-off was set to 400 eV, and the force threshold in structure optimization was 0.05 eV Å$^{-1}$. We used a large vacuum gap of 15 Å to eliminate the interactions between neighboring slabs. By adopting these calculation settings, the optimized lattice constant of CaO is 4.80 Å, which is in good agreement with the experimental value of 4.80 Å[61]. For the model construction, the $Ni_4$ cluster to simplify the theoretical calculation model cluster is widely used as a classic which could well reflect the characteristic of simulated nanoparticles in the DFT calculation[62–64]. Therefore, we built a p (4×4) surface slab containing five atom layers for CaO(100) and $Ni_4$ supported CaO(100) surfaces. The top four atom layers of these slabs were allowed to fully relax, while the bottom atom layer was kept fixed to mimic the bulk region. A $2 \times 2 \times 1$ k-point mesh was used in calculations of all these models. Calculation of the IR spectrum for the adsorbed of $CO_2$ and COOH species on the $Ni_4$–CaO(100) surfaces using the Born charges and the density functional perturbation theory (DFPT)[45,46]. For comparison, we also constructed the model Ni–CaO DFM with Ni 13-atom Ni clusters being supported on the CaO(100) substrate, namely $Ni_{13}$–CaO(100).

The transition states (TS) of surface reactions were located using a constrained optimization scheme and were verified when (i) all forces on the relaxed atoms vanish and (ii) the total energy is a maximum along the reaction coordination, but it is a minimum with respect to the rest of the degrees of freedom[65–67]. The adsorption energy of species $X$ on the surface ($E_{ads}(X)$) was calculated by the Eq. (4):

$$E_{ads}(X) = -(E_{X/slab} - E_{slab} - E_X) \tag{4}$$

where $E_{X/slab}$ is the calculated total energy of the adsorption system, while $E_{X/slab}$ and $E_X$ are calculated energies of the clean surface and the gas phase molecule $X$, respectively. Obviously, a positive value of $E_{ads}(X)$ indicates an exothermic adsorption process, and the more positive the $E_{ads}(X)$ is, the more strongly the adsorbate $X$ binds to the surface.

## Data availability

The data that support the findings of this study are available within the article and Supplementary Information or from the corresponding authors on reasonable request.

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

## Acknowledgements

We acknowledge the financial support from the National Natural Science Foundation of China (Nos. 22278126, 22250005, 22203030, 21878076, 21825301, and 92045303), the National Key Research & Development Program (2018YFA0208602), Intergovernmental International Science and Technology Innovation Cooperation Key Project (2021YFE0112800), and China Postdoctoral Science Foundation (2020M671020).

## Author contributions

J.H. and X.-Q.G. supervised the research. J.H. and B.S. conceived the ideas and designed the present work. B.S. performed dual-functional materials synthesis, characterization, and performance experiments. Z.-Q.W. and X.-Q.G. conducted the DFT calculations. H.-L.L., F.Q., and P.H. provided constructive suggestions. All authors contributed to the discussion and the paper writing.

## Competing interests

The authors declare no competing interests.
