## [Peer Review File · Nature Communications]

Synergistic Promotions between CO₂ Capture and in-situ Conversion on Ni-CaO Composite CatalystREVIEWER COMMENTS

Reviewer #1 (Remarks to the Author):

This manuscript deals with giving new insights on the Calcium-looping and dry reforming of methane (CaL@DRM) iCCC process, which is a promising cost-effective approach for carbon neutrality, on Ni-CaO/Al₂O₃ DFM. Based on conventional DFT calculations, the authors propose the most probable pathway to achieve the (CaL@DRM) process, revealing the essential role of an adequate balance the loading density and aggregation of Ni nanoparticles on porous CaO. Preparation of DFM follows methodology already described in recent papers by other authors (e.g. references 21, 22, 26, 28, 31). The prepared materials are characterized by PXRD, TEM, STEM, XPS, H₂-TPR, TPSR. The CO₂ capture and in-situ conversion experiments are performed in a fix bed reactor and products are analyzed d by on-line GC/NDIR techniques. The combination of DFT calculations with DFM properties and adsorption/DRM-reaction experiments allows the authors verifying the importance of the synergistic promotion of CO₂ capture/DRM-reaction by controlling structural variations of the Ni/CaO catal-sorbent (DFM).

In opinion of this reviewer, the manuscript is of interest for understanding the pathway of Ni/CaO DFM for the cyclic CO₂ capture and utilization for in-situ reforming of methane to yield the synthesis gas. I can consider DFT calculations as the most noteworthy results, significant and useful for the field. However, the used material (Ni-CaO)-DFM, as well as the cycled adsorption/reaction results related to the surface structure (3 samples varying 3 nickel loading) do not seem to me enough conceptually innovative related to the established knowledge. The combination of DFT with reaction experiments and FTIR analysis evidences the participation of different adsorbed intermediate species during reaction. In spite of different ad-species are defined (Figures S4-S6) and their existence probability, an unambiguous DRM reaction network cannot be outlined. In any case, novelty of concepts and approaches studied in the manuscript should be better stated.

In the following, I am suggesting some other considerations, in case they could help to authors for improving the manuscript.

1. For the model construction in DFT calculations, why the Ni₄ cluster has been chosen? How could results changed if different Ni aggregates are considered. See as an example, P. Wu et al. ACS Catalysis, 2019, 11, 10060-10069 (Cooperation of Ni and CaO at interface for CO₂ reforming of CH₄: A combined theoretical and experimental study).
2. Elemental analysis was performed using an inductively coupled plasma atomic emission spectrometer (ICP, Varia 710 ES). Which did the actual content of Ni result for 2.5, 5 and 10% nominal loadings? And the actual Ca content?
3. Second Figure 1 in the manuscript should be changed to Figure 2.
4. The fixed bed reactor is D10 mm × L150 mm, i.e. a volume of 11.8 cm³. If the bed is filled with 0.1 g Ni-CaO DFM, does it mean that density of DFM is 8.5×10⁻³ g cm⁻³? Or, is the DFM diluted in the bed? Something seems inconsistent.
5. For some authors in the cited literature, more recent references of importance for this manuscript are missed For example, refs. (26) and (28) should be completed with Journal of CO₂ Utilization, 2022, 58,

101922 (Tuning basicity of DFM ...). Similarly, ref. (31) should be completed with Sustainable Energy Fuels, 2020, 4, 5543-5549 (CO₂ green technologies in CO₂ capture and ...). 6. Some comparison of CO₂ and CH₄ conversion, and CO and H₂ production and ratio with other papers in the literature would be of interest to position the efficiency of the DFM and process. For example, comparison of results with those obtained in the latter paper mentioned in the previous item, with a the Ni/CaO catal-sorbent, and operating under similar operation conditions of temperature (650-700 °C), CO₂ and CH₄ concentrations (10% each), and total flow (40 vs. 50 ml min⁻¹). Some other papers for the (CaL@DRM) iCCC process under similar conditions can be found in the literature.

In conclusion, taking into account the address of Nature Communications to a rather general audience, in my opinion the manuscript, such as written, would fit better in a magazine more specialized in catalysis for CO₂ utilization, such as Journal of CO₂ Utilization, Topics in Catalysis, Sustainable Energy and Fuels, ...

Reviewer #3 (Remarks to the Author):

The paper “Synergistic Promotions between CO₂ Capture and in-situ Conversion” presents a combined experimental and theoretical work related to the integrated CO₂ capture and conversion (iCCC) technology, which deals here with Ni / calcite systems. The manuscript is well written, the figures well designed. One of the main goal of this study is to describe the interplay between so-called adsorption and catalytic sites in the investigated system. I am not convinced when reading this manuscript that the sites are different, and some conclusions are made too quickly. While the experimental part seems to be of good quality, the theoretical part presents big failures, from the choice of the size of cluster to the DFT settings to apply for such systems. Indeed, the experimental size of nickel clusters in this study is much bigger than the theoretical one, so the authors should resynthesize smaller nanoparticles (this is doable nowadays with specific control) and use a larger size in the DFT in the same time. I therefore recommend a major revision of this work before considering it for publication in Nature Comm.

1/ The part on DRM in the introduction should be a little bit developed, in particular with respect to some reviews recently published.

2/ I have a big concern about the choice of the Ni₄ cluster in the DFT calculations as the minimum particle size distribution is around 13 nm (line 191). As the interface between the cluster and the support plays a major role in the synergistic effect which is the main issue of this paper, I wonder what will happen using larger clusters: using 13 atoms instead of 4 would be a minimum (around 1 nm) on the DFT hand, or the authors could also synthesize smaller nanoparticles as well. In the current form, it is difficult to believe the conclusions drawn as the theoretical and experimental systems are very different. To help the authors to re-design their DFT study, I let here some references related to 13 atoms clusters:
<https://doi.org/10.1063/5.0007128>
<https://doi.org/10.1016/j.apsusc.2021.150790>

3/ There are major issues to address regarding DFT calculations. In fact, they have to be all redone using suitable settings. First of all, the authors have mentioned they have used spin-polarized calculations, which is a good point. What tag have they used? Do they systematically compared high spin and low spin configurations? This reference from the creators of VASP treating Ni systems could be cited.

[https://doi.org/10.1016/S0039-6028\(00\)00457-X](https://doi.org/10.1016/S0039-6028(00)00457-X)

4/ Have the authors applied DFT+U? this is mandatory to correctly describe Ni despite many published works dealing with small Ni clusters ignored this fact. I recommend the authors to be inspired by the following reference dealing with Ni cluster grafting,

<https://doi.org/10.1016/j.apsusc.2020.147422>

5/ The authors must include also dispersion corrections in their calculations as they want to compare adsorption and catalytic sites. A Grimme D2 correction should be enough (only 1% of cost of calculation). See and cite these two references when including D2 correction in VASP:

<http://dx.doi.org/10.1021/jp106469x>. <http://dx.doi.org/10.1002/jcc.20495>.

6/ Line 167: “characteristic peaks ... of the generated carbonates can be clearly seen in agreement with the DFT calculations” : this is not clear: have the authors computed the frequencies by DFT to compare with the experiments? I recommend to do it, because the bond involved here are quite strong and not quite subject to anharmonicity, so the comparison between experimental and theoretical frequencies will be easy. See one example involving fluorite (a calcium mineral like calcite) and fatty acids (similar head to CO₂) <https://doi.org/10.1016/j.jcis.2020.09.062>

This will give a plus to this manuscript, makes it easier to be publish in a good journal such as Nature Comm.

7/ Line 176 : please improve the readability of Fig. 3, some peaks are blurry.

8/ Line 214: The “excellent adsorption CO₂ capacities” should be compared with the literature.

9/ The title is too general, calcite and nickel should be specified there.

Dr Michael Badawi, Université de Lorraine, France

Response to Reviewers

Reviewer 1

“This manuscript deals with giving new insights on the Calcium-looping and dry reforming of methane (CaL@DRM) iCCC process, which is a promising cost-effective approach for carbon neutrality, on Ni-CaO/Al₂O₃ DFM. Based on conventional DFT calculations, the authors propose the most probable pathway to achieve the (CaL@DRM) process, revealing the essential role of an adequate balance the loading density and aggregation of Ni nanoparticles on porous CaO. dDFM follows methodology already described in recent papers by other authors (e.g. references 21, 22, 26, 28, 31). The prepared materials are characterized by PXRD, TEM, STEM, XPS, H₂-TPR, TPSR. The CO₂ capture and in-situ conversion experiments are performed in a fix bed reactor and products are analyzed d by on-line GC/NDIR techniques. The combination of DFT calculations with DFM properties and adsorption/DRM-reaction experiments allows the authors verifying the importance of the synergistic promotion of CO₂ capture/DRM-reaction by controlling structural variations of the Ni/CaO catal-sorbent (DFM). In opinion of this reviewer, the manuscript is of interest for understanding the pathway of Ni/CaO DFM for the cyclic CO₂ capture and utilization for in-situ reforming of methane to yield the synthesis gas. I can consider DTF calculations as the most noteworthy results, significant and useful for the field.”

Authors’ Response: We thank the reviewer for reading our manuscript so carefully and pointing out the key findings of our work. We also deeply appreciate the reviewer’s positive comments on our manuscript and recognition of the importance of DFT calculations in revealing catalytic mechanisms.

“However, the used material (Ni-CaO)-DFM, as well as the cycled adsorption/reaction results related to the surface structure (3 samples varying 3 nickel loading) do not seem to me enough conceptually innovative related to the established knowledge.”

Authors’ Response: We thank the reviewer for raising this concern. We are sorry that we did not explain the key progress made in our work clearly enough. In the current work, we learned from the DFT calculations that the adsorptive/catalytic interface structure is essential for the synergistic promotion between CO₂ capture and its *in-situ* conversion, and the reaction pathways of the *H-assistant *CO₂ conversion and the

*OH-assistant CH₄ dehydrogenation should be compatible with each other at the interface. Inspired by these results, we further tuned the Ni/CaO interface structure by varying the loading of Ni nanoparticles on the porous CaO substrate, and indeed, we verified that Ni-CaO-5 DFM with proper Ni loading (5 wt%) can give superior CO₂ capture capacity (11.91 mmol g⁻¹) and CO₂ conversion (94.1%), better than all the reported results of the conventional DRM under the similar reaction conditions (to the best of our knowledge).

Nevertheless, as suggested by the reviewer, we continued to verify such interfacial effect by further exploring how the size of Ni nanoparticles can affect the iCCC performance. By modifying the confinement space through varying the amount of soft template at the fixed loading of Ni (5 wt%) in the sol-gel process, a series of Ni-CaO-5(*d*) DFMs with the Ni particle size (*d*) ranging from 8.2 to 17.2 nm were obtained. As listed in **Table R1**, the dispersion degree of Ni increased with the decrease of Ni nanoparticle size, and hence, the smaller the Ni nanoparticle, the larger the interface can occur at Ni/CaO (**Figure R1**).

Table R1 (Table S4). Characteristic properties and activities of Ni-CaO-5 DFMs with different Ni sizes (*d*).

Sample	^a Ni loading (wt%)	^a Ca loading (wt%)	Ni HAADF-STEM mean particle size (nm)	^b Ni dispersion degree (%)	CO ₂ capture capacity (mmol g ⁻¹)	CO ₂ conversion (%)	CH ₄ conversion (%)
Ni-CaO-5(17.2)	5	66.3	17.2 ± 2.8	31	7.0	90.0	87.0
Ni-CaO-5(13.4)	5	66.8	13.4 ± 2.4	35	11.9	94.1	90.0
Ni-CaO-5(9.7)	5	66.5	9.7 ± 1.8	41	12.8	95.2	94.3
Ni-CaO-5(8.2)	5	66.4	8.2 ± 2.0	45	12.8	96.5	96

^a Ni and Ca loading determined by inductive coupled plasma emission spectrometer (ICP).

^b Ni dispersion obtained from CO chemisorption.

Figure R1 (Figure S20). TEM (left) and HAADF-STEM (right) results of (a) Ni-CaO-5(17.3); (b) Ni-CaO-5(13.4); (c) Ni-CaO-5(9.7) and (d) Ni-CaO-5(8.2).

In comparison with the original Ni-CaO-5(13.4), the smaller Ni nanoparticles of Ni-CaO-5(8.2) gave better adsorption and conversion performances, with the maximum CO₂ capture capacity of 12.8 mmol g⁻¹ and CO₂ and CH₄ conversion of 96.5% and 96.0%, respectively; on the contrary, the iCCC performance of the larger Ni nanoparticles of Ni-CaO-5(17.3) became much worse (**Figure R2**). This changing pattern of the iCCC performance with the Ni nanoparticle size provided further evidence that the Ni/CaO interface can facilitate the synergistic reaction pathways to promote the CO₂ capture and *in-situ* conversion.

Figure R2 (Figure S21). Effects of Ni particle size of Ni-CaO-5(*d*) DFMs on (a) CO₂ capture capacity and *in-situ* conversion efficiency under the optimized operating conditions and (b) specific yield of H₂ and CO (average yield per gram of the loaded Ni) along with the R_{H/C} in the syngas product at 650 °C.

It also needs to be mentioned that, through the sol-gel method, the sizes of the minimum Ni nanoparticles were limited at about 8-9 nm, and they gave very similar iCCC performance as Ni-CaO-5 (8.2) as well. In fact, to better explore the effect of the Ni nanoparticle size, we actually tested other synthesis approaches by using mesoporous silicon MCM-41 as a hard-templet. Through the co-impregnations of Ni and Ca species into MCM-41 and sequential calcination, the highly dispersed Ni nanoparticles in the size of only 2 nm at CaO was obtained (**Figure R3**). Such 5Ni-CaO/MCM-41(2) catalyst gave an extremely high CO₂ conversion of 99% with the H₂/CO molar ratio of nearly 1 (**Figure R4**), further demonstrating the crucial role of the Ni/CaO interface. However, due to the rather different synthesis approach of 5Ni-CaO/MCM-41(2) and its low CO₂ adsorption capacity (0.5 mmol g⁻¹), we did not include these results in the revised manuscript.

Figure R3. TEM image (left) and HAADF-STEM micrographs (right) of 5Ni-CaO/MCM-41(2).

Figure R4. Performance of CaL@DRM on the 5Ni-CaO/MCM-41(2) under 10 vol% CO₂/N₂ mixture flow for the capture step and 5 vol% CH₄/N₂ mixture flow for the conversion step.

Accordingly, in order to better explain the novel findings in this work, we have added **Figure R1** and **R2** as **Figure S20** and **S21**, and **Table R1** as **Table S4** in the revised Supporting Information and discussed the corresponding size effect in the revised manuscript (see **Page16, Line 263 – 273**).

“To elaborate upon the importance of interface structures for the iCCC performance, we regulated the sizes of Ni nanoparticles by varying the amount of soft template at the fixed loading of Ni (5 wt.%) in the sol-gel process. As the result of the different nanoconfined spaces constructed by the template, a series of Ni-CaO-5(*d*) DFMs with the Ni particle size (*d*) ranging from 8.2 to 17.2 nm were obtained (**Figure S20**). As listed in **Table S4**, the dispersion degree of Ni increased with the decrease of Ni nanoparticle size, and hence, the smaller the Ni nanoparticle, the larger the interface can occur at Ni/CaO. The Ni-CaO-5(*d*) DFMs also gave improved CO₂ adsorption and conversion with the decreased sizes of Ni nanoparticles. Specifically, Ni-CaO-5(8.2), with the smallest Ni particle size, exhibited the maximum CO₂ capture capacity of 12.8 mmol g⁻¹ and the best CO₂ and CH₄ conversion of 96.5% and 96.0%, respectively, together with the syngas production yield as high as 0.452 mmol s⁻¹ g_{Ni}⁻¹ (**Figure S21**). Such changing pattern of the iCCC performance with the Ni nanoparticle size provided further evidence that the enlarged Ni/CaO interface can facilitate the synergistic reaction pathways to promote the CaL@DRM reaction on the Ni-CaO DFM.”

“The combination of DFT with reaction experiments and FTIR analysis evidences the participation of different adsorbed intermediate species during reaction. In spite of

different ad-species are defined (Figures S4-S6) and their existence probability, an unambiguous DRM reaction network cannot be outlined.”

Authors’ Response: We thank the reviewer for raising this concern. We are sorry that the reaction network was not presented clearly enough. In the original manuscript, we explained the synergistic promotion of CaL@DRM pathway on the Ni-CaO interface as the following, “The captured $*CO_2$ can be readily converted with the help of $*H$ from the CH_4 dehydrogenation, while the overflow of $*OH$ produced in the above H-assisted $*CO_2$ conversion can interactively accelerate the CH_4 dehydrogenation on the $Ni_4-CaO(100)$ surface”. We believe the schematic illustration (Figure R5) had provided a general description of the reaction network involved in the above synergistic promotion processes. We also agree with the reviewer that more detailed descriptions of the pathways of CO_2 conversion and CH_4 dehydrogenation are necessary to support the outlined scheme.

Figure R5. Schematic illustration of reaction network for the CaL@DRM on the Ni₄-CaO(100) surface. The corresponding activation energy E_a (red) and reaction energy (black) in the unit of eV for each step are also included.

To do so, we first plotted the energy profiles of the CH₄ dehydrogenation (**Figure R6**), in which the calculated pathway in red is:

From these profiles, one can clearly see that the introduction of *OH produced from the *CO₂ conversion indeed changed the pathway of CH₄ dehydrogenation on the Ni sites at Ni₄-CaO(100) interface.

Figure R6. Calculated energy profiles of CH₄ dehydrogenation with the assistance of *OH species on the Ni₄-CaO(100) surface (red dashed line), the green dashed line represents the *CH₂ dissociation to *CH, and the brown dashed line represents the *CH₂OH dissociation to *CHOH.

In addition, we also calculated the energy profiles to illustrate how the produced *H from the above CH₄ dehydrogenation can overflow from the Ni site to the interface and involve in the *CO₂ dissociation (**Figure R7**). The comparison among the three different pathways of CO₂ dissociation, namely the *H-assisted COOH pathway, the *H-assisted HCOO pathway, and the direct CO₂ dissociation, clearly suggested that the first one is energetically the most favorable.

Figure R7. Calculated energy profiles of CO₂ adsorption and dissociation on the Ni₄-CaO(100) surface. The black dashed line represents the *H-assistant *HCOO pathway, the orange dashed line represent the direct CO₂ dissociation to *CO, and the blue dashed line represents the *H-assistant *COOH pathway.

Accordingly, in the revised manuscript, we have reconstructed the original **Figure 2** by combining the above three plots (see **Figure R8**), and rephrased the discussions of the corresponding parts (see **Page 9, Line 153 –168**) as the following:

“... From the calculated energetics of the above three different pathways, it can be concluded that the *COOH pathway is more favorable for the H-assisted *CO₂ dissociation (**Figure 2c**, inner ring). These results then indicate the synergistic promotion of CaL@DRM pathway on the Ni₄-CaO(100) surface, as the captured *CO₂ can be readily converted with the help of *H from the CH₄ dehydrogenation.

In addition, one may also notice that in the above direct dehydrogenation of CH₄, rather high energy barriers need to be overcome when the intermediate *CH₂ species undergoes deeper dehydrogenation (**Figure S7**). Interestingly, according to our calculations, the *OH produced from the conversion of *CO₂ can readily react with the *CH_x (x = 2, 1, 0) at the Ni sites to form *CH_xOH (**Figure S8 and S9**). It is found that *CH₂OH dissociation to the *CHOH species needs to overcome an energy barrier of 1.07 eV, which is 0.78 eV higher than that of *CH₂OH dissociation to *CH₂O (**Figure 2b**). So, the produced *OH would assist *CH₂ oxidation through the pathway *CH₂ + *OH → *CH₂OH → *CH₂O + *H → *CHO + *H → *CO + *H, with the energy barrier of 0.18 eV, 0.29 eV, 0.72 eV and 0.23 eV for each step, respectively (**Figure 2c**, outer ring). Therefore, the overflow of OH* produced in the H-assistant *CO₂ conversion can actually accelerate the CH₄ dehydrogenation on the Ni₄-CaO(100) surface, which

provides another strong evidence of the synergistic promotions.”

Figure R8 (Figure 3). DFT calculations and proposed reaction mechanism for CaL@DRM. (a) Calculated energy profiles of CO₂ adsorption and dissociation on the Ni₄-CaO(100) surface; the black dashed line represents the *H-assistant *HCOO pathway, the orange dashed line represent the direct CO₂ dissociation to *CO, and the blue dashed line represents the *H-assistant *COOH pathway. (b) Calculated energy profiles of CH₄ dehydrogenation with the assistance of *OH species on the Ni₄-CaO(100) surface (red dashed line); the green dashed line represents the *CH₂ dissociation to *CH, and the brown dashed line represents the *CH₂OH dissociation to *CHOH. (c) Schematic illustration of reaction network for the CaL@DRM on the Ni₄-CaO(100) surface. The corresponding activation energy E_a (red) and reaction energy (black) in the unit of eV for each step are also included.

“In any case, novelty of concepts and approaches studied in the manuscript should be better stated.”

Authors’ Response: We thank the reviewer for the valuable suggestion. In the above responses, we have explained more about the findings in this work and the corresponding revisions. Besides, following the reviewer’s suggestion, in the revised manuscript, we have also summed up these novelties in the last paragraph of the introduction part (see **Page 5, Line 82 - 88**).

“..., we demonstrated the synergistic promotion effects between CO₂ capture and *in-situ* conversion on the Ni-CaO DFM in a molecular way. Density functional theory

(DFT) calculations of the corresponding model catalyst with a Ni₄ cluster on the CaO(100) substrate disclosed the synergistic reaction pathways of the direct conversion of carbonate (activated CO₂) by DRM, which settles the debate about the role of captured CO₂ on conversion. More importantly, the mechanism of synergistic promotions provided the in-depth understanding of the long-sought consensus about the consecutive reactions involved in CaL@DRM and inspired the design and fabrication of DFMs with enriched interface between the catalytic site and adsorptive site. With the controlled loading density and the growth of Ni nanoparticles in the nanoconfined space of porous CaO substrate, such optimized Ni-CaO DFMs can give rise to high conversion efficiencies for both CO₂ and CH₄, superior to all the reported performance...”

“In the following, I am suggesting some other considerations, in case they could help to authors for improving the manuscript.

Question 1: *For the model construction in DFT calculations, why the Ni₄ cluster has been chosen?”*

Authors’ Response: We thank the reviewer for raising this concern. Honestly speaking, it is hard to find a definite criterion to determine the right model used in DFT calculations of a practical heterogeneous catalytic system. However, it can be generally accepted that a suitable size for saving computing time while with largely complete description of the real system should be important. In this work, Ni₄ cluster with a tetrahedral configuration on the CaO(100) surface was adopted as the “minimal” and acceptable model to simulate the Ni-CaO DFM, since various vertex, edge and even hollow sites can be obtained on this model to understand the catalytic performance of the Ni cluster as well as the its interface with the CaO support, which may also exhibit the synergistic promotion effects. In fact, similar Ni₄ cluster models have already been used in several other theoretical studies to successfully help reveal the characteristic roles of Ni nanocatalysts.^{1,2} Nevertheless, the model catalyst with a bigger Ni cluster was still built for testing calculations, and the detailed results are reported below.

Question 2: *“How could results changed if different Ni aggregates are considered. See as an example, P. Wu et al. ACS Catalysis, 2019, 11,10060-10069 (Cooperation of Ni and CaO at interface for CO₂ reforming of CH₄: A combined theoretical and experimental study).”*

Authors' Response: We thank the reviewer for this question. In order to deeply understand the effect of Ni aggregates, and in response to the concerns raised by the authors in the above two questions, we have constructed the Ni₁₃-CaO(100) model and calculated the adsorption of CO₂ and its dissociation by methane on it (**Figure R9a**)³⁻⁵. The calculated results showed that the CO₂ molecule can be adsorbed on the CaO surface by forming one C-O(CaO) bond and two (CO₂)O-Ca bonds, *i.e.*, a carbonate-like adsorbed species (*CO₂) occurs (**Table R2**). The corresponding calculated (exothermic) adsorption energy is about 1.98 eV (**Figure R9b**), stronger than that of the CO₂ on the supported Ni₁₃ cluster ($E_{\text{ads}} = 1.58$ eV, **Figure R9c**). These results indicated that the CO₂ prefers to be adsorbed on the CaO surface than the Ni₁₃ cluster, and this is consistent with the calculated relative stabilities of CO₂ at these two sites of Ni₄-CaO(100) in our original manuscript. In addition, we also investigated the cleavage of the C-H bond in methane on the Ni₁₃-CaO(100) surface (**Figure R9d-f**) and determined that the methane dissociation needs to overcome an energy barrier of 0.46 eV and is exothermic by 1.00 eV. Again, these results are largely consistent with those reported in our original manuscript (the methane dissociation barrier is 0.32 eV and the exothermic reaction energy is 1.36 eV on the Ni₄-CaO(100) surface). Therefore, though the calculated absolute values on the Ni₄- and Ni₁₃-CaO(100) models show some deviations, the relative stabilities and reactivities of the CO₂ and CH₄ molecules at different surface sites are unchanged, and therefore, the selection of Ni₄ cluster model is indeed reliable.

Figure R9 (Figure S4). (a) Calculated structures (left: side view; right: top view) of the Ni₁₃-CaO(100) surface. (b, c) Calculated structures (left: side view; right: top view) and adsorption energies of CO₂ on the Ni₁₃-CaO(100) surface. (d-f) Calculated structures (left: side view; right: top view) of CH₄ dissociation on the Ni₁₃-CaO(100) surface. Red: O; green: Ca; blue: Ni; grey: C; pink: H.

H; yellow: O of CO₂. This notation is used throughout the paper.

Question 3: “Elemental analysis was performed using an inductively coupled plasma atomic emission spectrometer (ICP, Varia 710 ES). Which did the actual content of Ni result for 2.5, 5 and 10% nominal loadings? And the actual Ca content?”

Authors’ Response: We thank the reviewer for raising this concern. In this work, we performed the elemental analysis using an inductively coupled plasma atomic emission spectrometer (ICP, Varia 710 ES). As shown in **Table R2**, the actual Ni content of Ni-CaO-2.5, Ni-CaO-5 and Ni-CaO-10 is 2.5 wt.%, 5.0 wt.% and 10 wt.%, respectively. Regarding the actual Ca content in Ni-CaO-x (x = 2.5, 5.0 and 10), we have added more information in the revised Supporting Information (see **Page S37, Table S1**) as following.

Table R2 (Table S1). Measurement results of element contents, crystallite sizes and porosities of Ni-CaO-x DFMs and pure CaO.

Sample	^a Ni(wt%)	^a Ca(wt%)	^b D _{Ni} (nm)	Ni dispersion (%)	^c D _{CaO} (nm)	^d S _{BET} (m ² g ⁻¹)	^e Pore volume (cm ³ g ⁻¹)
Ni-CaO-10	10	63.9	13.2±2.5	28	30	13	0.05
Ni-CaO-5	5	66.3	13.4±2.4	35	32	20	0.07
Ni-CaO-2.5	2.5	68.9	27.6±3.0	30	34	21	0.10
CaO	/	71.4	/	/	45	25	0.15

^a Measured by inductive coupled plasma emission spectrometer (ICP).

^b Estimated by size distributions of Ni nanoparticles in TEM images.

^c Average crystallite sizes calculated by $D_p = \frac{0.94\gamma}{\beta_{1/2} \cos \theta}$, based on different diffraction peaks at 2-theta.

^d BET surface areas.

^e Total pore volumes at a relative pressure (P/P₀) of 0.99.

Question 4: “Second Figure 1 in the manuscript should be changed to Figure 2.”

Authors’ Response: We thank the reviewer for reading our manuscript so carefully and pointing out this error. We have carefully checked all figure numbers and captions throughout the manuscript and made corrections in the revised manuscript.

Question 5: “The fixed bed reactor is $\Phi 10 \text{ mm} \times L150 \text{ mm}$, i.e. a volume of 11.8 cm^3 . If the bed is filled with 0.1 g Ni-CaO DFM , does it mean that density of DFM is $8.5 \times 10^{-3} \text{ g cm}^{-3}$? Or, is the DFM diluted in the bed? Something seems inconsistent.”

Authors’ Response: We thank the reviewer for raising this concern. The fixed bed reactor used in this work has the size of $\Phi 10 \text{ mm} \times L150 \text{ mm}$. Meanwhile, the height of the packed bed of 0.1 g Ni-CaO DFM powder was about 10 mm , i.e., a volume of $\sim 0.785 \text{ cm}^3$. So, the packing density of DFM is actually 0.13 g/cm^3 . The corresponding explanation has now been corrected in the revised SI (see **Page S6, Line 104-105**).

“...approximately $0.1 \text{ g Ni-CaO-x DFM}$ was added into a quartz reaction tube ($\Phi 10 \text{ mm} \times 150 \text{ mm}$) and packed with the height of about 10 mm , ...”

Question 6: “For some authors in the cited literature, more recent references of importance for this manuscript are missed. For example, refs. (26) and (28) should be completed with *Journal of CO₂ Utilization*, 2022, 58, 101922 (Tuning basicity of DFM ...). Similarly, ref. (31) should be completed with *Sustainable Energy Fuels*, 2020, 4, 5543-5549 (CO₂ green technologies in CO₂ capture and ...).”

Authors’ Response: We deeply thank the reviewer for pointing out this issue. We have updated the relevant references in the revised manuscript as the following:

“...and it is not clear if the adsorption site can solely accommodate the CO₂ and its catalytic conversion can only occur at other positions,²³⁻²⁶ or the captured CO₂ can be also activated at the adsorption site for direct reaction with the incoming H species produced on the nearby catalytic sites.²⁷⁻³⁰...”

[25] Bermejo-López, A., Pereda-Ayo, B., Onrubia-Calvo, J.A., González-Marcos, J.A. & González-Velasco, J.R. Tuning basicity of dual function materials widens operation temperature window for efficient CO₂ adsorption and hydrogenation to CH₄. *J. CO₂ Utiliz.* **58**, 101922 (2022).

[29] Jo, S.B., et al. CO₂ green technologies in CO₂ capture and direct utilization processes: methanation, reverse water-gas shift, and dry reforming of methane. *Sustain. Energy Fuels* **4**, 5543-5549 (2020).”

Question 7: “Some comparison of CO₂ and CH₄ conversion, and CO and H₂ production and ratio with other papers in the literature would be of interest to position the

efficiency of the DFM and process. For example, comparison of results with those obtained in the latter paper mentioned in the previous item, with a the Ni/CaO catal-sorbent, and operating under similar operation conditions of temperature (650-700 o C), CO₂ and CH₄ concentrations (10% each), and total flow (40 vs. 50 ml min⁻¹). Some other papers for the (CaL@DRM) iCCC process under similar conditions can be found in the literature.”

Authors’ Response: We thank the reviewer for these valuable suggestions. As suggested by the reviewer, we have added relevant references of the conventional DRM processes involving direct CO₂ conversion by CH₄⁷⁻¹⁶ and the processes catalyzed by CaO based DFMs operating under similar conditions⁵⁻⁶ for comparison (**Table R3**).

Among all the reported CaO-based DFMs for the CaL@DRM process, Ni-CaO-5(8.2) DFM produced by the modified sol-gel method showed the highest CO₂ conversion and CH₄ conversion of 96.5% and 96.0%, respectively. More significantly, the H₂/CO molar ratio in the elution profile was nearly 1, suggesting that the Ni-CaO-5 DFM had a much better resistance to carbon deposition compared with the Ni-Ca@Zr and Ni-CaO catal@sorbent.

Accordingly, these comparisons have been summarized in **Table S5** in the revised Supporting Information (**Page S41**), together with the corresponding discussions in the revised manuscript (see **Page 16, Line 275 - 279**), “Compared with the catalysts reported in other studies (**Table S5**), the Ni-CaO-5(8.2) DFM produced by the modified sol-gel method showed the highest CO₂ conversion and CH₄ conversion of 96.5% and 96.0%, respectively. More significantly, the H₂/CO molar ratio in the elution profile was nearly 1, suggesting that the Ni-CaO-5 DFM had a much better resistance to carbon deposition compared with the Ni-Ca@Zr and Ni-CaO catal@sorbent.^{29,52,”}

Table R3 (Table S5). Comparisons of the CaL@DRM iCCC processes and conventional DRM processes reported in the literatures.

Catalyst	Gas composition	WHSV	Temperature (C°)	CH ₄ conversion (%)	CO ₂ conversion (%)	H ₂ /CO molar ratio (R _{H/C})	References
Ni-CaO-5 DFM	Capture step: 10 vol.% CO ₂ /N ₂ ;	30 L g ⁻¹ h ⁻¹ (50 ml/min)	650	96.0	96.5	1.02	This work
	Conversion step: 5vol.% CH ₄ /N ₂						
Ni-CaO catal-sorbent	Capture step: 10 vol.% CO ₂ /N ₂ ;	12 L g ⁻¹ h ⁻¹ (40 ml/min)	700	/	83.8	6.53	6
	Conversion step: 10vol.% CH ₄ /N ₂						
Ni-Ca@Zr	Capture step: 5 vol.% CO ₂ /N ₂ ;	36 L g ⁻¹ h ⁻¹ (30 ml/min)	720	40	45	0.8	7
	Conversion step: 8 vol.% CH ₄ /N ₂						
Ni/SiO ₂ -E	CH ₄ /CO ₂ /N ₂ = 3:3:4	30 L g ⁻¹ h ⁻¹	500	7.9	17.9	0.5	8
NiFe/Al ₂ O ₃	CH ₄ /CO ₂ = 1:1	12 L g ⁻¹ h ⁻¹	550	26.6	37.8	0.67	9
0.3PdNi/MCM-41	CH ₄ /CO ₂ /N ₂ = 1:1:3	120 L g ⁻¹ h ⁻¹	550	37	50	0.8	10
Ni-CaO/MCM-41	CH ₄ /CO ₂ /Ar = 1:1:1	143 L g ⁻¹ h ⁻¹	650	20	40	0.5	5
15%Ni@S-1	CH ₄ /CO ₂ = 1:1	100 L g ⁻¹ h ⁻¹	650	40	42	0.6	11
Ni ₃ Fe ₁ Cu ₁ -MA	CH ₄ /CO ₂ /N ₂ = 2:2:1	432 L g ⁻¹ h ⁻¹	650	15	35	0.5	12
Ni/MCF	CH ₄ /CO ₂ /Ar = 9:9:2	36 L g ⁻¹ h ⁻¹	650	60	70	1.0	13
Ni/MgO	CH ₄ /CO ₂ /Ar = 1:1:8	30 L g ⁻¹ h ⁻¹	700	≥37	≥50	0.78	14
Ni/SiO ₂	CH ₄ /CO ₂ /N ₂ = 9:9:2	12 L g ⁻¹ h ⁻¹	800	40	62	0.7	15
NiMo/MgO	CH ₄ /CO ₂ /He = 1:1:8	60 L g ⁻¹ h ⁻¹	800	100	100	1	16

“In conclusion, taking into account the address of Nature Communications to a rather general audience, in my opinion the manuscript, such as written, would fit better in a magazine more specialized in catalysis for CO₂ utilization, such as Journal of CO₂ Utilization, Topics in Catalysis, Sustainable Energy and Fuels, .../”

Authors’ Response: We thank the reviewer for reading our manuscript so carefully and recognizing the key contributions of this work. We agree with the reviewer that the CaL@DRM process we proposed is a special approach to convert CH₄ and the CO₂ from the high-temperature flue gas to syngas. However, we must also emphasize the novelty as well as the significance of this work for CO₂ capture and utilization through the most meaningful high-temperature iCCC technology. By investigating the detailed CaL@DRM process with combined experimental and theoretical methods, for the first time, we demonstrated the synergistic promotion effects between CO₂ capture and *in-situ* conversion at molecular level. Our findings of the direct conversion of CaCO₃ by the dry reforming of methane also settled the debate about the role of captured CO₂ on conversion. More importantly, the mechanism of synergistic promotions provides not only in-depth understanding of the long-sought consensus about the consecutive reactions at molecular level, it may also inspire future research for more efficient iCCC processes. Moreover, with the global aim for carbon neutrality, the development of such dual function materials provides a solution to for CO₂ reduction in high carbon emission industries, especially the steel, cement, petrochemistry industries, which can be already seen from the comparison (added in the revised manuscript as suggested by the reviewer) of the catalytic performances between the DFMs proposed in this work and those from the literatures. Therefore, we truly believe that that our manuscript involves enough novelty and significance and will attract wide attention in the fields of C1 chemistry, surface science, material science, and thus merits publication in *Nature Communications*.

Reviewer #2

“The present work describes the ICCC (Integrated CO₂ Capture and Conversion) technology through an experimental approach supported by theoretical calculations by means DFT (Density Functional Theory). Specifically, the proposed ICCC technology (CaL@DRM), involves the integration of the Calcium Looping (CaL) process (repeated cycles of CO₂ absorption on CaO solid sorbent and regeneration) and the dry reforming of methane (DRM) process with CO₂ previously absorbed on the solid sorbent and catalyzed by Ni metallic supported on the same solid sorbent, namely, a dual functional material (DFM). The topic of the present work is particularly attractive in the field of research of global warming due to anthropogenic CO₂ emissions and energy saving of industrial processes. The manuscript is well written and the experiments are well organized and described, as well as the results are thoroughly discussed and conclusions well drawn. The scientific soundness and novelty of the present paper makes it suitability for publication in Nature Communications journal.”

Authors’ Response: We sincerely thank the reviewer for reading our manuscript so carefully and pointing out the key results of our work. We also thank the reviewer for the positive and encouraging comments on our work.

“In the following are the changes should be done.

1). Pg. 5, Line 92-93: “..., CaO can give rise to steady CH₄ elution curves without any consumption below 800°C;”. Above 800°C some H₂ product can be observed from Fig. 1(b) due to thermal cracking of CH₄ (CH₄ → C + 2H₂). The same can be seen in Fig. S1(c). Explain.”

Authors’ Response: We thank the reviewer for raising this concern. As the thermal cracking of CH₄ (CH₄ → C + 2H₂, ΔH_{298K} = +74.6 kJ/mol) is an endothermic reaction, it will be more favorable to occur under higher temperatures. Thus, the thermal cracking of CH₄ was determined in this work when the temperature was above 800 °C even without the help of any catalyst. As the result, CH₄-TPSR curves of the pure CaO sample and the CaO after CO₂ pre-adsorption (CaCO₃), together with the H₂ formation, can be detected above 800 °C (**Figure R10**).

To explain these results more clearly, we have rephrased the discussion part related to **Figure 1b** and **Figure S1c** in the revised manuscript (**Page 6, Line 101 - 106**) as the following: **“As the thermal cracking of CH₄ (CH₄ → C + 2H₂, ΔH_{298K} = +74.6 kJ/mol)**

is an endothermic reaction, it will be more favorable to occur under higher temperatures. Thus, the thermal cracking of CH₄ was determined in this work when the temperature was high enough even without the help of any catalyst, giving rise to the CH₄-TPSR curves of the pure CaO sample and the CaO after CO₂ pre-adsorption (CaCO₃), together with the H₂ formation, above 800 °C (**Figure 1b** and **Figure S1c**.)”

Figure R10. The performances of the CH₄ temperature-programmed surface reactions (CH₄-TPSR) on the (a) pure CaO and (b) CaO after CO₂ pre-adsorption.

“2). Pg. 7, Line 112: In Figure 1(b) denote Ni-CaO-x (x?) because you have prepared Ni-CaO-x (x=2.5, 5, and 10). The same is for Fig. S1 of file 377720_0_supp_6692946_r41647.”

Authors’ Response: We thank the reviewer for the valuable suggestion. The sample mentioned by the reviewer is Ni-CaO-10. Now, we have specified it in the revised captions of Figure 1(b) and Figure S1 in the revised manuscript.

“3) Pg. 9, Line 153: incorrect figure number, write “Figure 2” instead of “Figure 1”.”

Authors’ Response: We thank the reviewer for reading our manuscript so carefully and pointing out the incorrect figure number. We are also very sorry about this mistake, and we have now corrected it in the revised manuscript.

“4). Pg. 10, Line 156: separate the figure label from the text.”

Authors' Response: We thank the reviewer for this suggestion. In the revised manuscript, we have separated the **Figure 1** caption from the text.

“5). Pg. 14, Line 226: Write “Raman” instead of “Ramon”.”

Authors' Response: We deeply thank the reviewer for reading our manuscript so carefully and pointing this error. We are sorry for this typo error and have now carefully checked the language throughout the manuscript.

“6). Pg.15, Line 240: “... mat reactor.”. What does it mean?”

Authors' Response: We thank the reviewer for raising this concern, and we are sorry that we did not explain clearly in this part of the original manuscript. We have now revised this sentence in the revised manuscript (see **Page 17, Line 280-281**) as the following. “In addition, to realize the consecutive reactions of the exothermic CO₂ capture and the endothermic conversion in the same reactor for the iCCC processes, the reaction temperature plays a pivotal role^{7,59}.”

“File name: 377720_0_supp_6692946_r4l647

7). Pg.S3, Line 40: after: ” microscopy (SEM) was conducted on a Helios G4 UC SEM-FIB (15 kV)” add dot.”

Authors' Response: Again, we deeply thank the reviewer for reading our manuscript so carefully. We have added the dot (full stop) at the end of the sentence in the revised Supporting Information (see **Page S3, Line 47**). “Scanning electron microscopy (SEM) was conducted on a Helios G4 UC SEM-FIB (15 kV).”, and carefully checked the format throughout the manuscript, including the Supporting Information.

“8). Pg. S3, Line 40: Check number of Micromeritics ASAP. Is perhaps 2020 and not 3020?”

Authors' Response: We deeply thank the reviewer for pointing out this error. The number of Micromeritics ASAP is 2020 and we have corrected it as “ASAP-2020” in the revised Supporting Information (see **Page S3, Line 54**).

“9). Pg. S6, Line 113: $R_{H/C}$ is molar ratio as have written in Table S3.”

Authors’ Response: We thank the reviewer for this question. The reviewer is right that the $R_{H/C}$ is the molar ratio of H_2/CO in the obtained syngas. We have added the note for this symbol below **Table S3** in the revised Supporting Information (see **Page S39**) as:

Table S3. The performance of Ni-CaO-x DFMs with different Ni loading in the CaL@DRM processes at 650 °C.

Sample	CO ₂ capacity (mmol g ⁻¹)	CO yield (mmol s ⁻¹ g _{Ni} ⁻¹)	H ₂ yield (mmol s ⁻¹ g _{Ni} ⁻¹)	^a $R_{H/C}$	Carbon balance (%)	CO ₂ conversion (%)	CH ₄ conversion (%)
Ni-CaO-2.5	12.10	0.170	0.150	0.88	64.00	86.40	84.30
Ni-CaO-5	11.91	0.172	0.182	1.06	96.50	94.10	90.00
Ni-CaO-10	11.58	0.064	0.080	1.25	87.90	95.40	92.70
CaO	12.60	0	0	/	/	/	/

^a $R_{H/C}$ is the molar ratio of H_2/CO in the obtained syngas.

“10). Pg. S6, Line 116: Write “...into the mixture of 10 vol.% CO₂, 10 vol.% CH₄ balanced with N₂.” instead of “...into the mixture of 10 vol.% CO₂, 10 vol.% CH₄ and N₂””

Authors’ Response: We thank the reviewer for this suggestion. We have made the corresponding correction in the revised Supporting Information (see **Page S7, Line 135**) as: “... and the feeding gas was changed into the mixture of 10 vol.% CO₂ and 10 vol.% CH₄ balanced with N₂.”

“11). Pg. S27, Table S2: Check the value of $R_{H/C}$ of sample Ni-CaO-2.5. Is perhaps 0.88 value instead of 0.84?”

Authors’ Response: We thank the reviewer for reading our manuscript so carefully and pointing out this error. We must also apologize for such mistake. The value of $R_{H/C}$ of Ni-CaO-2.5 is indeed 0.88 and we have corrected it in the revised manuscript and Supporting Information.

“Furthermore

Could the authors better explain why they carried out the conventional DRM test under 10 vol.% CO₂ + 10 vol.% CH₄ - N₂, whereas CaL@DRM test was conducted under 5vol.% CH₄-N₂ as well the produced intermediates characterized by an in-situ diffuse reflectance infrared Fourier transform spectroscopy was performed in 25 vol.% CH₄-He? Why did they use different gas concentrations? The performances of DFMs could be different owing to the different operating conditions.”

Authors’ Response: We thank the reviewer for raising these concerns. According to the stoichiometric numbers of the DRM reaction ($\text{CO}_2 + \text{CH}_4 = 2\text{CO} + 2\text{H}_2$), the molar ratio of CH₄/CO₂ in the feedstock gas should be 1. Therefore, we carried out the conventional DRM test by using the mixture of 10 vol.% CO₂ and 10 vol.% CH₄ balanced with N₂.

For the CaL@DRM process, it involves the consecutive CO₂ capture step and the conversion step of the adsorbed CO₂ by CH₄. After the enrichment of CO₂ from the diluted flue gas (~10 vol%), the concentration of CH₄ in the following conversion step would correlate with the amount of adsorbed CO₂ on the Ni-CaO DFM and the catalytic conversion rate. We optimized the concentration of CH₄ by comparing the conversion results of CaL@DRM tests under the same adsorption condition but with different concentrations of 5 vol% and 10 vol % CH₄ balanced with N₂. It turned out that the CO₂ capture capacity was the same (11.91 mmol g⁻¹), while in the conversion step, the H₂/CO molar ratio of product was 2.4 under 10 vol% CH₄, higher than the 1.06 under 5 vol% CH₄. Moreover, excessive unreacted CH₄ was detected in the elution curves. Therefore, 5 vol% CH₄ was well matched with the conversion rate of the reduction of the adsorbed CO₂ (**Figure R11a**), while 10 vol% CH₄ was actually excessive, and itself decomposed to give H₂ to make the H₂/CO molar ratio higher in the products (**Figure R11b**). This is the reason why we chose the mixture of 5 vol% CH₄ balanced with N₂ as the reactant in the conversion step.

Figure R11. Molar flow rate of the effluent gases in one cycle of CaL@DRM on the Ni-CaO-5 under the 10 vol% CO₂/N₂ feed gas for the capture step but (a) 5 vol% and (b) 10 vol% CH₄/N₂ for the conversion step.

For the intermediate characterization, it is usually very difficult to conduct the *in-situ* DRIFTS test. We therefore increased the concentration of CH₄ to 25 vol% in order to obtain the prominent signal of the intermediates, and to make it more consistent with the CaL@DRM test, we have also repeated the *in-situ* DRIFTS test with the feed gases of 10 vol% CO₂/N₂ for the capture step and 5 vol% CH₄/N₂ for the conversion step (**Figure R12**). The results turned out to be the same as those obtained under the 25 vol% CH₄ feed gas, and in the revised manuscript, we have replaced the original **Figure 3** with the figure below.

Figure R12 (Figure 3c and 3d). *In-situ* DRIFTS spectra of CaL@DRM iCCC processes on the Ni-CaO-5. (a) The CO₂ adsorption stage in the atmosphere of 10 vol.% CO₂ balanced with N₂, and (b) the *in-situ* CO₂ conversion stage in the atmosphere of 5 vol.% CH₄ balanced with N₂.

Reviewer #3

*“The paper “Synergistic Promotions between CO₂ Capture and in-situ Conversion” presents a combined experimental and theoretical work related to the integrated CO₂ capture and conversion (iCCC) technology, which deals here with Ni / calcite systems. The manuscript is well written, the figures well designed. One of the main goal of this study is to describe the interplay between so-called adsorption and catalytic sites in the investigated system. I am not convinced when reading this manuscript that the sites are different, and some conclusions are made too quickly. While the experimental part seems to be of good quality, the theoretical part presents big failures, from the choice of the size of cluster to the DFT settings to apply for such systems. Indeed, the experimental size of nickel clusters in this study is much bigger than the theoretical one, so the authors should resynthesize smaller nanoparticles (this is doable nowadays with specific control) and use a larger size in the DFT in the same time. I therefore recommend a **major revision** of this work before considering it for publication in Nature Comm.”*

Authors’ Response: We deeply appreciate the positive comments from the reviewer and the recognition of the novelty and importance of our work. We also thank the reviewer for raising the concerns regarding the sizes of the catalysts in our experimental and theoretical studies.

As suggested by the reviewer, we further fabricated smaller Ni nanoparticles to demonstrate the size effect on the iCCC performance. To do so, we fixed the loading amount of Ni at 5 wt% and increased the amount of soft template in the sol-gel synthesis process to make the Ni more dispersed. Accordingly, smaller Ni nanoparticles in Ni-CaO-5(*d*) DFMs with the size (*d*) decreasing from 13.4 nm to 9.7 nm and 8.2 nm were obtained. These results have also been illustrated in **Table R1** and **Figure R1** as we replied to the comments from Reviewer 1, and for better clarity, they are given below too.

Table R2 (Table S4). Characteristic properties of Ni-CaO-5 DFM with different Ni sizes.

Sample	^a Ni loading (wt%)	^a Ca loading (wt%)	Ni HAADF-STEM mean particle size (nm)	^b Ni dispersion degree (%)	CO ₂ capture capacity (mmol g ⁻¹)	CO ₂ conversion (%)	CH ₄ conversion (%)
Ni-CaO-5(17.2)	5	66.3	17.2 ± 2.8	31	7.0	90.0	87.0
Ni-CaO-5(13.4)	5	66.8	13.4 ± 2.4	35	11.9	94.1	90.0

Ni-CaO-5(9.7)	5	66.5	9.7 ± 1.8	41	12.8	95.2	94.3
Ni-CaO-5(8.2)	5	66.4	8.2 ± 2.0	45	12.8	96.5	96

^a Ni and Ca loading determined by inductive coupled plasma emission spectrometer (ICP).

^b Ni dispersion obtained from CO chemisorption.

Figure R1 (Figure S20). TEM image (left) and HAADF-STEM (right) results of (a) Ni-CaO-5(17.3); (b) Ni-CaO-5(13.4); (c) Ni-CaO-5(9.7) and (d) Ni-CaO-5(8.2).

In comparison with the original Ni-CaO-5(13.4), the smaller Ni nanoparticles of Ni-CaO-5(8.2) gave better adsorption and conversion performances, with the maximum CO₂ capture capacity of 12.8 mmol g⁻¹ and CO₂ and CH₄ conversion of 96.5% and 96.0%, respectively; on the contrary, the iCCC performance of the larger Ni nanoparticles of Ni-CaO-5(17.3) became much worse (**Figure R2**). This changing pattern of the iCCC performance with the Ni nanoparticle size provided further evidence that the Ni/CaO interface can facilitate the synergistic reaction pathways to promote the CO₂ capture and *in-situ* conversion.

Figure R2 (Figure S21). Effects of Ni particle size of Ni-CaO-5(*d*) DFMs on the (a) CO₂ capture capacity and *in-situ* conversion efficiency under the optimized operating conditions, and (b) specific yield of H₂ and CO (average yield per gram of the loaded Ni) along with the $R_{H/C}$ in the syngas product at 650 °C.

It also needs to be mentioned that, through the sol-gel method, the sizes of the minimum Ni nanoparticles were limited at about 8-9 nm, and they gave very similar iCCC performance as Ni-CaO-5(8.2) as well. In fact, to better explore the effect of the Ni nanoparticle size, we actually tested other synthesis approaches by using mesoporous silicon MCM-41 as a hard-templet. Through the co-impregnations of Ni and Ca species into MCM-41 and sequential calcination, the highly dispersed Ni nanoparticles in the size of only 2 nm at CaO was obtained (**Figure R3**). Such 5Ni-CaO/MCM-41(2) catalyst gave an extremely high CO₂ conversion of 99% with the H₂/CO molar ratio of nearly 1 (**Figure R4**), further demonstrating the crucial role of the Ni/CaO interface. However, due to the different synthesis approach of 5Ni-CaO/MCM-41(2) and its low CO₂ adsorption capacity (0.5 mmol g⁻¹), we did not include these results in the revised manuscript.

Figure R3. TEM image (left) and HAADF-STEM micrographs (right) of 5Ni-CaO/MCM-41(2).

Figure R4. Performance of CaL@DRM on the 5Ni-CaO/MCM-41(2) under 10 vol% CO₂/N₂ mixture flow for the capture step and 5 vol% CH₄/N₂ mixture flow for the conversion step.

Accordingly, in order to better explain the novel findings in this work, we have added **Figure R1 and R2** as **Figure S20** and **Figure S21**, and **Table R1** as **Table S4** in the revised Supporting Information and discussed the corresponding size effect in the revised manuscript (see **Page16, Line 263 – 273**).

“To elaborate upon the importance of interface structures for the iCCC performance, we regulated the sizes of Ni nanoparticles by varying the amount of soft template at the fixed loading of Ni (5 wt.%) in the sol-gel process. As the result of the different nanoconfined spaces constructed by the template, a series of Ni-CaO-5(*d*) DFMs with the Ni particle size (*d*) ranging from 8.2 to 17.2 nm were obtained (**Figure S20**). As listed in **Table S4**, the dispersion degree of Ni increased with the decrease of Ni nanoparticle size, and hence, the smaller the Ni nanoparticle, the larger the interface can occur at Ni/CaO. The Ni-CaO-5(*d*) DFMs also gave improved CO₂ adsorption and conversion with the decreased sizes of Ni nanoparticles. Specifically, Ni-CaO-5(8.2), with the smallest Ni particle size, exhibited the maximum CO₂ capture capacity of 12.8 mmol g⁻¹ and the best CO₂ and CH₄ conversion of 96.5% and 96.0%, respectively, together with the syngas production yield as high as 0.452 mmol s⁻¹ g_{Ni}⁻¹ (**Figure S21**). Such changing pattern of the iCCC performance with the Ni nanoparticle size provided further evidence that the enlarged Ni/CaO interface can facilitate the synergistic reaction pathways to promote the CaL@DRM reaction on the Ni-CaO DFM.”

“Question 1: The part on DRM in the introduction should be a little bit developed, in

particular with respect to some reviews recently published.”

Authors’ Response: We thank the reviewer for this suggestion. One short paragraph about the DRM, including recent progresses and viewpoints, has been added in the introduction section of the revised manuscript (on **Page 4, Line 70 - 76**):

“... At the same time, the dry reforming of methane (DRM) is a very useful reaction, which can convert the two major greenhouse gases of CO₂ and CH₄ into valuable syngas with equimolar H₂ and CO. The challenges in the development of DRM mainly come from the high energy consumptions due to the strong endothermic nature of this reaction and the corresponding high reaction temperature (> 800 °C).^{32,33} Moreover, it also suffers from the catalyst deactivation caused by coking.^{34,35} Currently, most studies focus on developing efficient DRM catalysts for the activation of the two reactants, while the strategies to take advantage of their coupled reactions are still limited. ...

[32] Buelens, L.C., Galvita, V.V., Poelman, H., Detavernier, C. & Marin, G.B. Super-dry reforming of methane intensifies CO₂ utilization via Le Chatelier’s principle. *Science* **354**, 449-452 (2016).

[33] Song, Y., et al. Dry reforming of methane by stable Ni–Mo nanocatalysts on single-crystalline MgO. *Science* **367**, 777-781 (2020).

[34] Akri, M., et al. Atomically dispersed nickel as coke-resistant active sites for methane dry reforming. *Nat. Commun.* **10**, 5181 (2019).

[35] Kurlov, A., et al. Exploiting two-dimensional morphology of molybdenum oxycarbide to enable efficient catalytic dry reforming of methane. *Nat. Commun.* **11**, 4920 (2020). ”

“Question 2: I have a big concern about the choice of the Ni₄ cluster in the DFT calculations as the minimum particle size distribution is around 13 nm (line 191). As the interface between the cluster and the support plays a major role in the synergistic effect which is the main issue of this paper, I wonder what will happen using larger clusters: using 13 atoms instead of 4 would be a minimum (around 1 nm) on the DFT hand, or the authors could also synthesize smaller nanoparticles as well. In the current form, it is difficult to believe the conclusions drawn as the theoretical and experimental systems are very different. To help the authors to re-design their DFT study, I let here some references related to 13 atoms clusters:

<https://doi.org/10.1063/5.0007128>

<https://doi.org/10.1016/j.apsusc.2021.150790>”

Authors' Response: We thank the reviewer for this valuable suggestion. Following the reviewer's suggestion (as well as the similar one from Reviewer 1), we have prepared more DFM samples of Ni-CaO-5(*d*) with much smaller Ni nanoparticles. As explained above, the size effect on the CaL@DRM iCCC performance was found to be consistent with each other, which can be attributed to the fact that the enlarged Ni/CaO interface can facilitate the synergistic reaction pathways. Also, for the size of the calculated Ni cluster, as we have explained in our response to the Reviewer 1's comment, the Ni₄ could still be the suitable one as it involves the characteristic Ni sites and Ni/CaO interfacial sites.

Nevertheless, according to the reviewers' suggestions, we have also constructed the model Ni-CaO DFM with 13-atom Ni clusters being supported on the CaO(100) substrate, namely Ni₁₃-CaO (**Figure R9a**).⁵ The calculations of the CO₂ adsorption and the methane dissociation on the Ni₁₃-CaO(100) surfaces were carried out^{3,4}. The results showed the same adsorption performance as those on the Ni₄-CaO that one CO₂ prefers to be adsorbed on the CaO surface by forming one C-O(CaO) bond and two (CO₂)O-Ca bonds, *i.e.*, a carbonate-like adsorbed species (*CO₂) occurs. The corresponding calculated exothermic adsorption energy is about 1.98 eV (**Figure R9b**), stronger than that of the CO₂ on the Ni₁₃ cluster ($E_{\text{ads}} = 1.58$ eV, **Figure R9c**). These conclusions are consistent with that in our original manuscript (the adsorption of CO₂ is exothermic by about 1.21 eV on the CaO surface and stronger than that of the CO₂ on the Ni₄ cluster ($E_{\text{ads}} = 1.19$ eV)). Notably, we can find that the adsorption of CO₂ on the Ni₁₃-CaO(100) surface is stronger than that on the Ni₄-CaO(100) surface, which may be caused by the richer surface and interface structures of the Ni₁₃-CaO systems for the adsorbed CO₂ to achieve more comfortable interactions. In addition, we also investigated the cleavage of the C-H bond in methane at Ni₁₃-CaO(100) (**Figures R9d - f**), and the results showed that the methane dissociation needs to overcome an energy barrier of 0.46 eV and gives an exothermic reaction energy of 1.00 eV. These results are also consistent with those in our original manuscript (the dissociation of methane needs overcome energy barrier of 0.32 eV and gives an exothermic reaction energy of 1.36 eV at Ni₄-CaO(100)). We have added **Figure R9** as **Figure S4** and the references mentioned above have been cited as [39, 40] in the revised manuscript (see **Page 8, Line 135**) as the following "...The calculated adsorption energy (E_{ads}) is 1.21 eV, slightly stronger than that of the CO₂ on the Ni₄ cluster ($E_{\text{ads}} = 1.19$ eV), and similar results can be also obtained from supported Ni₁₃ clusters (**Figure S4**, see **SI** for details).^{39,40} ...

[39] Gueddida, S., Lebègue, S., Pasc, A., Dufour, A. & Badawi, M. Ab initio investigation of the adsorption of phenolic compounds, CO, and H₂O over metallic cluster/silica catalysts for hydrodeoxygenation process. *Appl. Surf. Sci.* **567**, 150790 (2021).

[40] Gueddida, S., Badawi, M. & Lebègue, S. Grafting of iron on amorphous silica surfaces from ab initio calculations. *J. Chem. Phys.* **152**, 214706 (2020).”

Figure R9 (Figure S4). (a) Calculated structures (left: side view; right: top view) of Ni₁₃-CaO(100) surface. (b-c) Calculated structures (left: side view; right: top view) and adsorption energies of CO₂ on the Ni₁₃-CaO (100) surface. (d-f) Calculated structures (left: side view; right: top view) of CH₄ dissociation on the Ni₁₃-CaO(100) surface. Red: O; green: Ca; blue: Ni; grey: C; pink: H; yellow: O of CO₂. This notation is used throughout the paper.

“Question 3: There are major issues to address regarding DFT calculations. In fact, they have to be all redone using suitable settings. First of all, the authors have mentioned they have used spin-polarized calculations, which is a good point. What tag have they used? Do they systematically compared high spin and low spin configurations? This reference from the creators of VASP treating Ni systems could be cited. [https://doi.org/10.1016/S0039-6028\(00\)00457-X](https://doi.org/10.1016/S0039-6028(00)00457-X)”

Authors’ Response: We thank the reviewer for raising this concern. As the reviewer pointed out, it is indeed important to consider the spin-polarization in the calculations of the Ni related systems.¹⁷ In our original manuscript, we did not systematically compare the effects of high and low spin configurations on calculated results of the integrated CO₂ capture and conversion reactions. Actually, we used the tag (ISPIN = 2) in the spin-polarized calculations and did not set the MAGMOM value for each atom. So, these VASP calculations will determine the spin states automatically (see **Table R4**).

The calculated results showed that the total magnetic moments of the Ni₄-CaO(100) surface before and after CO₂ adsorption are both 2 μ B (namely, the average magnetic moment per Ni atom is 0.50 μ B), which is slightly smaller than the experimental value of \sim 0.6 μ B.¹⁸ Therefore, the calculation settings in this work should still give largely reliable results for these Ni-based catalysts. Besides, we have also added the reference mentioned above as [3] in the revised Supporting Information (**Page S7, Line 139**).

“... All spin-polarized DFT calculations were carried out using the Vienna Ab-initio Simulation Package (VASP).^{2,3}...”

[3] Kresse, G. & Hafner, J. First-principles study of the adsorption of atomic H on Ni (111), (100) and (110). *Surf. Sci.* **459**, 287-302 (2000).”

Table R4. Calculated magnetic moments (M) of Ni₄-CaO(100) with and without adsorbed CO₂.

	Ni ₄ -CaO(100)	Ni ₄ -CaO(100)-CO ₂ ads1	Ni ₄ -CaO(100)-CO ₂ ads2	Ni ₄ -CaO(100)-CO ₂ ads3	Ni ₄ -CaO(100)-CO ₂ ads4
M / μ B	2	2	2	2	2

“Question 4: Have the authors applied DFT+U? this is mandatory to correctly describe Ni despite many published works dealing with small Ni clusters ignored this fact. I recommend the authors to be inspired by the following reference dealing with Ni cluster grafting,

<https://doi.org/10.1016/j.apsusc.2020.147422>”

Authors’ Response: We thank the reviewer for this valuable suggestion. As the reviewer mentioned, the influence of Hubbard U correction may need to be considered in the calculation of Ni based materials.⁴ However, in the original manuscript, we did not consider the influence of U value on the calculations of the synergistic CaL@DRM iCCC processes, and in fact, the calculation method we have adopted was consistent with those used in many other studies.^{2,19} Nevertheless, following the reviewer’s suggestion, we have conducted testing calculations with the Hubbard U correction being applied to the Ni 3d states to better describe the atomic and electronic structures of the Ni₄-CaO(100) surfaces, where the U_d = 3.00 eV and J = 0.90 eV were used,⁴ respectively. The calculated results showed that the overall trend of CO₂ adsorption at different sites of the Ni₄-CaO(100) surface was consistent with that obtained without U correction (**Figure R13**). For example, the CO₂ still prefers to be adsorbed on the CaO surface by forming one C-O(CaO) bond and two (CO₂)O-Ca bonds, *i.e.*, a carbonate-

like adsorbed species (*CO₂) occurs; and the corresponding adsorption energy is 1.14 eV (**Figure R13e**), stronger than that of the CO₂ on the Ni₄ cluster (0.92 eV, **Figure R13f**), both of which are also close to those obtained without U correction. Therefore, we still believe that the results we reported in our manuscript, especially those regarding the relative stabilities and reactivities, are largely reliable. We have added **Figure R13** as **Figure S5** in the revised Supporting Information (**Page S19**) and the reference mentioned above by the reviewer has been cited as [45] in the revised manuscript (see **Page 8, Line 139**).

“...and Hubbard U correction⁴⁵ were also considered in the calculations of Ni based materials, and the calculated results showed that the overall trend of CO₂ adsorption at different sites of the Ni₄-CaO(100) surface were consistent with that obtained without Grimme D2 or U corrections (Figures S5 and S6, see SI for details). ...”.

[45] Gueddida, S., Lebègue, S. & Badawi, M. Interaction between transition metals (Co, Ni, and Cu) systems and amorphous silica surfaces: A DFT investigation. *Appl. Surf. Sci.* **533**, 147422 (2020).

Figure R13 (Figure S5). Calculated structures (left: side view; right: top view) and adsorption energies of CO₂ on the Ni₄-CaO(100) surface obtained without (a-d) and with (e-h) the on-site Coulomb interaction correction.

“**Question 5:** The authors must include also dispersion corrections in their calculations as they want to compare adsorption and catalytic sites. A Grimme D2 correction should be enough (only 1% of cost of calculation). See and cite these two references when including D2 correction in VASP:

<http://dx.doi.org/10.1021/jp106469x>.

<http://dx.doi.org/10.1002/jcc.20495>.”

Authors’ Response: We again thank the reviewer for this valuable suggestion. We

agree with the reviewer that the dispersion corrections would be important in the calculations of the adsorptions of reactants. Following the reviewer's suggestion, we have performed the testing calculations with the dispersion corrections by using the DFT-D2 method.^{20,21} The calculated results showed that the adsorption energies of CO₂ on the Ni₄-CaO(100) surface increase by ~0.2 eV under the dispersion corrections, but the overall trend of CO₂ adsorption strengths at different site of the Ni₄-CaO(100) surface was largely unchanged (**Figure R14**). The CO₂ also prefers to be adsorbed on the CaO surface by forming one C-O(CaO) bond and two (CO₂)O-Ca bonds, *i.e.*, a carbonate-like adsorbed species (*CO₂) occurs. Therefore, we still believe that our calculated results without considering dispersion corrections are largely reliable. We have added **Figure R14** as **Figure S6** in the revised Supporting Information (see **Page S20**) and the references mentioned above have been cited as [43,44] in the revised manuscript (see **Page 8, Line 138**) as following “**Notably, the dispersion corrections by using the DFT-D2 method**^{43,44}...

[43] Grimme, S. Semiempirical GGA-type density functional constructed with a long-range dispersion correction. *J. Comput. Chem.* **27**, 1787-1799 (2006).

[44] Bučko, T., Hafner, J., Lebègue, S. & Ángyán, J.G. Improved description of the structure of molecular and layered crystals: Ab initio DFT calculations with van der waals corrections. *J. Phys. Chem. A* **114**, 11814-11824 (2010).”

Figure R14 (Figure S6). Calculated structures (left: side view; right: top view) and adsorption energies of CO₂ on the Ni₄-CaO(100) surface without (a-d) and with (e-f) the Grimme D2 correction.

“Question 6: Line 167: “characteristic peaks ... of the generated carbonates can be clearly seen in agreement with the DFT calculations”: this is not clear: have the authors computed the frequencies by DFT to compare with the experiments? I recommend to do it, because the bond involved here are quite strong and not quite subject to anharmonicity, so the comparison between experimental and theoretical

frequencies will be easy. See one example involving fluorite (a calcium mineral like calcite) and fatty acids (similar head to CO₂).

<https://doi.org/10.1016/j.jcis.2020.09.062>

This will give a plus to this manuscript, makes it easier to be publish in a good journal such as Nature Comm.”

Authors’ Response: We deeply appreciate this valuable suggestion from the reviewer. As recommended, we computed the frequencies of the adsorbed CO₂ and *COOH intermediate on the Ni₄-CaO(100) surface by using the density-functional perturbation theory (DFPT)^{20,22} (**Figure R15a, b**). As the DFT calculations showed that a single CO₂ preferred to form two O-Ca bonds to generate a carbonate-like adsorbed species on the CaO(100) terrace, we used this carbonate mode for adsorbed CO₂. The calculated results showed the occurrence of asymmetric stretching frequencies at 1740 and 905 cm⁻¹ (**Figure R15a**), well consistent with the characteristic peaks of 1789 and 876 cm⁻¹ of the generated carbonate (*CO₂) on the CaO surface determined experimentally in the CO₂ capture stage (**Figure R15c**). Moreover, the calculated asymmetric stretching of the monodentate adsorption mode of *COOH on the CaO(100) surface is at 1362 cm⁻¹ (**Figure R15b**), actually verifying that the newly formed peak at 1350 cm⁻¹ in the experimental conversion stage can be ascribed to this key intermediate (**Figure R15d**).

By combining the experimental *in-situ* DRIFTS spectra and DFPT calculation results, we reconstructed **Figure 3 (Figure R15)**, and the references mentioned above have been cited as [49] in the revised manuscript (see **Page 10, Line 182-193**). The corresponding discussions have been revised the as the following.

“We further verified this synergistic mechanism for the CaL@DRM processes through the *in-situ* diffuse reflectance infrared Fourier transform spectroscopy (DRIFTS) and the density-functional perturbation theory (DFPT) calculation.⁴⁹⁻⁵¹ In the CO₂ capture stage, the optimized carbonate-like *CO₂ species on the CaO(100) surface obtained from DFT calculations was used, and the calculated asymmetric stretching frequencies are at 1740 and 905 cm⁻¹ (**Figure 3a**). Correspondingly, the increasing peaks of the carbonate (*CO₂) on the CaO surface at 1789 and 876 cm⁻¹ and of the calcite (CaCO₃) at 1540 and 2504 cm⁻¹ can be clearly observed in the DRIFTS spectra (**Figure 3c**)⁵². After the N₂ elution, the gas was switched into CH₄ for the following CO₂ conversion stage. According to the DFPT calculation, the asymmetric stretching of the adsorbed monodentate *COOH on the CaO(100) gives the peak at 1362 cm⁻¹ (**Figure 3b**). Accordingly, a newly formed peak at 1350 cm⁻¹ in the DRIFTS spectra

can be also seen and verified to be the key intermediate of *COOH (Figure 3d).⁵³ These results indeed further confirmed the proposed *H-assisted pathway for the *CO₂ conversion.

[49] Foucaud, Y., et al. Adsorption mechanisms of fatty acids on fluorite unraveled by infrared spectroscopy and first-principles calculations. *J. Colloid Interface Sci.* **583**, 692-703 (2021).”

Figure R15 (Figure 3). Characteristic spectra of CaL@DRM iCCC processes on the Ni-CaO. The DFPT calculated peaks from the asymmetric stretching of the adsorbed (a) CO₂ (*CO₂) and (b) COOH group (*COOH). *In-situ* DRIFTS spectra during (c) the CO₂ adsorption stage in the atmosphere of 10 vol.% CO₂ balanced with N₂, and (d) the CO₂ conversion stage in the atmosphere of 5 vol.% CH₄ balanced with N₂ of the CaL@DRM iCCC process on the Ni-CaO surface.

“**Question 7:** Line 176: please improve the readability of Fig. 3, some peaks are blurry.”

Authors’ Response: We thank the reviewer for this suggestion. We have reconstructed the **Figure 3** (as **Figure R15**) in the revised manuscript to improve the quality and reliability of the spectra evidence.

“Question 8: Line 214: The “excellent adsorption CO₂ capacities” should be compared with the literature.”

Authors’ Response: We thank the reviewer for this suggestion. As illustrated in **Table R4**, with the flow rate of the flue gas at 50 ml min⁻¹, Ni-CaO-5 showed excellent performance with the adsorption capacity being kept as high as 12.8 mmol g⁻¹ and the CO₂ conversion being maintained at about 96.5% at 650 °C, which are indeed superior in comparison with the state-of-the-art studies of CaO-based DFMs. We have added the **Table R4** as **Table S2** in the revised Supporting Information (see Page S37).

Table R4 (Table S2). Summary of CaO-based dual function materials and their iCCC performances.

DFMs	Reaction	Operation condition		Flow rate (ml min ⁻¹)	CO ₂ adsorption capacity (mol kg ⁻¹)	CO ₂ conv. (%)	Ref.
		Adsorption	Reaction				
Ni-CaO/Al ₂ O ₃	Methanation	320 °C 9.5% CO ₂ /N ₂	320 °C 10% H ₂ /N ₂	100	0.31	46	23
Ru-CaO/Al ₂ O ₃	Methanation	320°C, 10% CO ₂ /N ₂	320°C, 10% H ₂ /N ₂	17.4	0.68	96	24
Ni-CaO-Ce	RWGS	650–750 °C 15% CO ₂ /N ₂	650–750 °C 5% H ₂ /N ₂	100	14.1	52	25
Fe ₅ Co ₅ Mg ₁₀ CaO	RWGS	650°C, 10% CO ₂ /N ₂	650°C, 100% H ₂	50	9.2	90	26
Ni-(K-Ca)/γ-Al ₂ O ₃	DER	650 °C 10% CO ₂ /N ₂	650 °C 5% C ₂ H ₆ /N ₂	30	0.99	65	27
Ni-(Na-Ca)/γ-Al ₂ O ₃	DER	650 °C 10% CO ₂ /N ₂	650 °C 5% C ₂ H ₆ /N ₂	30	0.63	75	27
Ni-CaO catal-sorbent	DRM	700 °C 10% CO ₂ , 10 % H ₂ O/N ₂	700 °C 10% CH ₄ /N ₂	40	14.8	83.8	6
Ni-Ca@Zr	DRM	720 °C, 5 % CO ₂ /Ar	720 °C 8% CH ₄ /Ar	30	5	45	7
Ni-CaO-5	DRM	650 °C, 10 % CO ₂ / N ₂	650 °C, 5 % CH ₄ /N ₂	50	12.8	96.5	This work

“Question 9: The title is too general, calcite and nickel should be specified there.”

Authors’ Response: We thank the reviewer for this suggestion. The title has now been modified as “**Synergistic Promotions between CO₂ Capture and *in-situ* Conversion on Ni-CaO Composite Catalyst**”.

“Dr Michael Badawi, Université de Lorraine, France”

Authors’ Response: We thank *Dr. Badawi* for spending time reviewing our manuscript and providing the valuable suggestions and comments to help us improve the quality of the manuscript.

References

1. Zuo, Z., *et al.* Dry reforming of methane on single-site Ni/MgO catalysts: importance of site confinement. *ACS Catal.* **8**, 9821-9835 (2018).
2. Liu, H., *et al.* CH₄ dissociation on the perfect and defective MgO(001) supported Ni₄. *Fuel* **123**, 285-292 (2014).
3. Gueddida, S., Badawi, M. & Lebègue, S. Grafting of iron on amorphous silica surfaces from ab initio calculations. *J. Chem. Phys.* **152**, 214706 (2020).
4. Gueddida, S., Lebègue, S. & Badawi, M. Interaction between transition metals (Co, Ni, and Cu) systems and amorphous silica surfaces: A DFT investigation. *Appl. Surf. Sci.* **533**, 147422 (2020).
5. Wu, P., *et al.* Cooperation of Ni and CaO at interface for CO₂ reforming of CH₄: A combined theoretical and experimental study. *ACS Catal.* **9**, 10060-10069 (2019).
6. Jo, S.B., *et al.* CO₂ green technologies in CO₂ capture and direct utilization processes: methanation, reverse water-gas shift, and dry reforming of methane. *Sustain. Energy Fuels* **4**, 5543-5549 (2020).
7. Hu, J., Hongmanorom, P., Galvita, V.V., Li, Z. & Kawi, S. Bifunctional Ni-Ca based material for integrated CO₂ capture and conversion via calcium-looping dry reforming. *Appl. Catal. B* **284**, 119734 (2021).
8. Wang, F., Han, K., Xu, L., Yu, H. & Shi, W. Ni/SiO₂ catalyst prepared by strong electrostatic adsorption for a low-temperature methane dry reforming reaction. *Ind. Eng. Chem. Res.* **60**, 3324-3333 (2021).
9. Song, Z., *et al.* Improved effect of Fe on the stable NiFe/Al₂O₃ catalyst in low-temperature dry reforming of methane. *Ind. Eng. Chem. Res.* **59**, 17250-17258 (2020).
10. Damyanova, S., *et al.* MCM-41 supported PdNi catalysts for dry reforming of methane. *Appl. Catal. B Environ.* **92**, 250-261 (2009).
11. Liu, Y., *et al.* Embedding high loading and uniform Ni nanoparticles into silicalite-1 zeolite for dry reforming of methane. *Appl. Catal. B* **307**, 121202 (2022).
12. Jin, F., *et al.* Stable trimetallic NiFeCu catalysts with high carbon resistance for dry reforming of methane. *ChemPlusChem* **85**, 1120-1128 (2020).
13. Daoura, O., *et al.* Mesocellular silica foam-based Ni catalysts for dry reforming of CH₄ (by CO₂). *J. CO₂ Utiliz.* **24**, 112-119 (2018).
14. Cheng, F., Duan, X. & Xie, K. Dry Reforming of CH₄ /CO₂ by stable Ni

- nanocrystals on porous single-crystalline MgO monoliths at reduced temperature. *Angew. Chem. Int. Ed. Engl.* (2021).
15. Han, J.W., Park, J.S., Choi, M.S. & Lee, H. Uncoupling the size and support effects of Ni catalysts for dry reforming of methane. *Appl.Catal. B* **203**, 625-632 (2017).
 16. Song, Y., *et al.* Dry reforming of methane by stable Ni–Mo nanocatalysts on single-crystalline MgO. *Science* **367**, 777-781 (2020).
 17. Kresse, G. & Hafner, J. First-principles study of the adsorption of atomic H on Ni (111), (100) and (110). *Surf. Sci.* **459**, 287-302 (2000).
 18. C. Kittel. Introduction to solid state physics, 5th Edition; Wiley: New York, 1976.
 19. Guo, Y., Feng, J. & Li, W. Effect of the Ni size on CH₄/CO₂ reforming over Ni/MgO catalyst: A DFT study. *Chin. J. Chem. Eng.* **25**, 1442-1448 (2017).
 20. Bučko, T., Hafner, J., Lebègue, S. & Ángyán, J.G. Improved description of the structure of molecular and layered crystals: Ab initio DFT calculations with van der waals corrections. *The Journal of Physical Chemistry A* **114**, 11814-11824 (2010).
 21. Grimme, S. Semiempirical GGA-type density functional constructed with a long-range dispersion correction. *J. Comput. Chem.* **27**, 1787-1799 (2006).
 22. Baroni, S., de Gironcoli, S., Dal Corso, A. & Giannozzi, P. Phonons and related crystal properties from density-functional perturbation theory. *Reviews of Modern Physics* **73**, 515-562 (2001).
 23. Chai, K.H., Leong, L.K., Wong, D.S.H., Tsai, D.H. & Sethupathi, S. Effect of CO₂ adsorbents on the Ni-based dual-function materials for CO₂ capturing and in situ methanation. *J. Chin. Chem. Soc.* **67**, 998-1008 (2020).
 24. Arellano-Treviño, M.A., He, Z., Libby, M.C. & Farrauto, R.J. Catalysts and adsorbents for CO₂ capture and conversion with dual function materials: Limitations of Ni-containing DFMs for flue gas applications. *J. CO₂ Utiliz.* **31**, 143-151 (2019).
 25. Sun, H., *et al.* Dual functional catalytic materials of Ni over Ce-modified CaO sorbents for integrated CO₂ capture and conversion. *Appl.Catal. B* **244**, 63-75 (2019).
 26. Shao, B., *et al.* Heterojunction-redox catalysts of Fe_xCo_yMg₁₀CaO for high-temperature CO₂ capture and in situ conversion in the context of green manufacturing. *Energy Environ.Sci.* **14**, 2291-2301 (2021).

27. Al-Mamoori, A., Rownaghi, A.A. & Rezaei, F. Combined capture and utilization of CO₂ for syngas production over dual-function materials. *ACS Sustainable Chem. Eng.* **6**, 13551-13561 (2018).

REVIEWERS' COMMENTS

Reviewer #1 (Remarks to the Author):

The main concerns that I highlighted in my original review have been consistently considered by the authors and incorporated in the revised version of the manuscript. My comments and suggestions for improving the manuscript have also been considered and adequately responded to, with corresponding incorporations in the revised version.

In my opinion, the manuscript has been enriched over the previous version and I believe it now deserves publication in Nature Communication. I suggest acceptance.

Reviewer #2 (Remarks to the Author):

This manuscript is well revised. It can be accepted for publication

Reviewer #3 (Remarks to the Author):

The authors have adressed seriously all my concerns and also the comments of all reviewer. The manuscript has been significantly improved and should be published as it is in Nature Communciations.

Response to Reviewers

Reviewer 1

“The main concerns that I highlighted in my original review have been consistently considered by the authors and incorporated in the revised version of the manuscript. My comments and suggestions for improving the manuscript have also been considered and adequately responded to, with corresponding incorporations in the revised version. In my opinion, the manuscript has been enriched over the previous version and I believe it now deserves publication in Nature Communication. I suggest acceptance.”

Authors’ Response: We thank the reviewer for his/her help in making it possible for our work to be published.

Reviewer 2

“This manuscript is well revised. It can be accepted for publication”

Authors’ Response: We thank the reviewer for his/her help in making it possible for our work to be published.

Reviewer 3

“The authors have addressed seriously all my concerns and also the comments of all reviewer. The manuscript has been significantly improved and should be published as it is in Nature Communications.”

Authors’ Response: We thank the reviewer for his/her help in making it possible for our work to be published.